# Physiologically Based Pharmacokinetic Modeling of Transdermal Selegiline and Its Metabolites for the Evaluation of Disposition Differences between Healthy and Special Populations

**DOI:** 10.3390/pharmaceutics12100942

**Published:** 2020-09-30

**Authors:** Santosh Kumar Puttrevu, Sumit Arora, Sebastian Polak, Nikunj Kumar Patel

**Affiliations:** 1Certara UK Limited, Simcyp Division, Level 2-Acero, 1 Concourse Way, Sheffield S1 2BJ, UK; Sumit.Arora@certara.com (S.A.); sebastian.polak@certara.com (S.P.); nikunjkumar.patel@certara.com (N.K.P.); 2Pharmacoepidemiology and Pharmacoeconomics Unit, Faculty of Pharmacy, Jagiellonian University Medical College, Medyczna 9, str., 30-688 Krakow, Poland

**Keywords:** selegiline, transdermal, multi-phase multi-layer mechanistic dermal absorption (mpml-mechderma) model, parent and metabolite PBPK model, special populations

## Abstract

A physiologically based pharmacokinetic (PBPK) model of selegiline (SEL), and its metabolites, was developed in silico to evaluate the disposition differences between healthy and special populations. SEL is metabolized to methamphetamine (MAP) and desmethyl selegiline (DMS) by several CYP enzymes. CYP2D6 metabolizes the conversion of MAP to amphetamine (AMP), while CYP2B6 and CYP3A4 predominantly mediate the conversion of DMS to AMP. The overall prediction error in simulated PK, using the developed PBPK model, was within 0.5–1.5-fold after intravenous and transdermal dosing in healthy and elderly populations. Simulation results generated in the special populations demonstrated that a decrease in cardiac output is a potential covariate that affects the SEL exposure in renally impaired (RI) and hepatic impaired (HI) subjects. A decrease in CYP2D6 levels increased the systemic exposure of MAP. DMS exposure increased due to a reduction in the abundance of CYP2B6 and CYP3A4 in RI and HI subjects. In addition, an increase in the exposure of the primary metabolites decreased the exposure of AMP. No significant difference between the adult and adolescent populations, in terms of PK, were observed. The current PBPK model predictions indicate that subjects with HI or RI may require closer clinical monitoring to identify any untoward effects associated with the administration of transdermal SEL patch.

## 1. Introduction

R-(−) Selegiline (SEL) is a selective irreversible mono amine oxidase-B (MAO-B) inhibitor approved for the treatment of major depressive disorder (MDD), it is also used in oral adjunct therapy with Levodopa for the treatment of late-stage Parkinson disease [1,2]. Orally administered SEL undergoes extensive first pass metabolism (4% oral bioavailability) [3], which led to the development of the selegiline transdermal patch system (STS). STS is available at 6 mg, 9 mg, and 12 mg daily doses with corresponding patch sizes of 20 mg/20 cm^2^, 30 mg/20 cm^2^, and 40 mg/20 cm^2^. The patch formulation is specifically prescribed for the treatment of MDD in patients ≥12 years of age [4]. The dermal route of administration of this MAO inhibitor avoids the tyramine-free diet (reaction to cheese) restrictions for the 6 mg/24 h dose, but its administration is restricted for the 9 mg and 12 mg/24 h doses due to the lack of sufficient safety data [4]. Suicidality is the major risk factor associated with the administration of anti-depressant drugs such as SEL; therefore, patients undergoing MDD therapy must balance this therapeutic risk against the benefits [5]. Short-term clinical studies revealed that patients above 24 years of age are at a lower suicidal risk compared to children, adolescents and young adults [6]. However, patients of all age groups, who are under chronic repetitive anti-depressant therapy, should undergo closer clinical monitoring for their suicidal tendencies [5]. 

Conversely, the clinical pharmacokinetic (PK) aspects of SEL have been extensively reviewed in the literature [3,7,8,9,10,11]. Upon once-daily dermal administration, about 30% of the drug content is delivered systemically over 24 h [5]. Biotransformation of SEL predominantly occurs in the liver, where it primarily metabolizes to methamphetamine (MAP) and desmethyl selegiline (DMS) and subsequently into the secondary metabolite amphetamine (AMP) [12,13]. MAP and DMS are the major and minor primary metabolites, which are formed by depropargylation and demethylation, respectively [8,13]. The conversion of SEL to its primary metabolites is mediated by a multitude of CYP enzymes in the liver [14]. CYP2D6 metabolizes the conversion of MAP to AMP, while CYP2B6 and CYP3A4 (with a minor contribution from CYP2A6) mediate the conversion of DMS to AMP [5,15]. In terms of plasma protein binding, SEL is highly bound; in contrast, MAP and AMP exhibit low plasma protein binding (predominantly to albumin) [16,17]. SEL binds preferentially to macroglobulin and to some extent albumin [18,19]. The CYP enzymes (CYPs 2B6, 2D6, and 3A4) involved in the metabolism of MAP and DMS to AMP have variable turnover rates and polymorphisms; this diversity adds complexity to the disposition and systemic exposure of the metabolites [11,20,21,22,23,24,25,26].

PBPK modeling defines the pharmacokinetics of drugs by delineating the physiological (system specific data) and physicochemical characteristics of the drug(s) and drug product(s) [27]. In the bottom-up approach used in this study, virtual human populations were built based on the physiological parameters (age, sex, and ethnicity) and incorporated the genetic makeup of relevant enzymes, transporter proteins, etc. In many cases, the system specific parameters exhibited non-linear and non-monotonic relationships. These complex covariate relationships influence the absorption, distribution, metabolism, and excretion (ADME) aspects of the drug [27]. The pharmacokinetics of oral SEL has been investigated in renal and hepatic impaired subjects and was found to be associated with a 6- and 18-fold increase, respectively, in SEL exposure compared to heathy adults [28]. The pharmacokinetics of SEL was associated with a higher inter-individual variability in healthy, renally, and hepatic impaired subjects [28]. In the present study, a PBPK model based approach was utilized to evaluate the pharmacokinetic differences of SEL, and its metabolites, when administered as a transdermal patch. Therefore, the objective was to develop and verify a comprehensive PBPK model for SEL and its metabolites in healthy and geriatric populations and to further apply the model to predict the PK in virtual special populations namely adolescent, renally impaired, and hepatic impaired populations after transdermal dosing. An additional objective was to identify the key physiological covariates that are responsible for the disposition differences observed between healthy and special populations to inform whether these special populations require closer clinical monitoring of the parent compound and the formation of its metabolites after transdermal dosing of this anti-depressant therapy.

## 2. Methods

### 2.1. Pharmacokinetic Data

Intravenous and transdermal clinical PK data was collected by performing an extensive literature search in PubMed, Google Scholar, and the United States Food and Drug Administration New Drug Application (NDA) package. The reports that provided parent and metabolite PK data were used for model verification, the details of which are summarized in Appendix A. Only one clinical PK study for intravenous infusion dose was available in the literature [3]. Transdermal PK studies after single and multiple dosage regimens were reported in both healthy adult and geriatric populations [3,9,10,11]. Two studies reported in elderly male and female subjects by Barret et al., one that used a dose of 9.15 mg/10 cm^2^ and another that used dose proportionality (0.5, 1, and 1.5 mg/cm^2^) in elderly males; females were not considered for the model verification. This was due to the high variability reported in the observed data (>80% CV in pharmacokinetic parameters) in the dose proportionality study. The reason for the Barret et al. 1996 study at 9.15 mg/10 cm^2^ not being considered was due to missing data values (MDV) and the presence of concentrations below the lower limit of quantification in some cases [9,11].

### 2.2. Simcyp Simulator Setup

PBPK modeling and simulations were performed using the Simcyp population based Simulator version 18, release 2 (Certara, Sheffield, UK). The virtual populations for healthy volunteers (Sim-Healthy Volunteers [29]), geriatric populations (Sim-Geriatric NEC [30,31]), moderate and severe renally impaired populations (Sim-Renal GFR 30–60 and Sim-Renal GFR < 30 [32]), and moderate and severe hepatic impaired populations (Sim-Cirrhosis Child Pugh-B and Sim-Cirrhosis Child Pugh-C [33]) were used for subject sampling. The macroglobulin protein concentrations in healthy (males: 3.95 μM; 17.7% CV and females: 4.47 μM; 14.4% CV) [34], renally impaired (males: 7.27 μM; 37.3% CV and females: 8.21 μM; 21.9% CV) [35], and hepatic impaired population (males: 5.488 μM; 23.88% CV and females: 5.23 μM; 19.2% CV) [35] were defined based on the literature reports. PBPK simulations were performed in 10 virtual trials with the number of subjects (n) in each trial equal to the corresponding clinical study. Hence, the total number of virtual subjects in each PBPK simulation were 10 × n. The simulation trials were matched with the clinically reported trial design (number of subjects, age, gender, etc.) as shown in Appendix A. The default values for CYP enzyme abundance and the phenotype frequencies (extensive metabolizers: EM, intermediate metabolizers: IM, poor metabolizers:PM, and ultrarapid metabolizers: UM) of different populations that were used for the simulations are summarized in Appendix A. 

### 2.3. PBPK Model Workflow

The current study reports a comprehensive parent and metabolite whole body PBPK model to explain the disposition characteristics of the parent drug (SEL) to the primary metabolites (MAP, DMS) and further to a secondary metabolite (AMP). The PBPK model workflow and schematic representation of SEL and its metabolites (MAP, DMS and AMP) are presented in Figure 1a and Figure 1b, respectively. The model input parameters of the parent drug and its metabolites are summarized in Table 1. The clinical PK studies utilized in the model development and verification are summarized in Appendix A. Parameter sensitivity analysis was performed to identify the sensitive parameters, which were subsequently optimized by a parameter estimation method to match the observed data. Initial model development was performed using intravenous data in healthy volunteers, followed by single transdermal PK in the healthy subjects. The intrinsic clearance (CL_int_) values of DMS were optimized using data retrieved from the intravenous PK study reported by Azzaro et al. [3]. The first order release rate constant for the patch formulation was optimized using data provided by a single dose transdermal PK study in healthy subjects [3]. The optimized model was verified across different clinically reported PK studies that constituted different dosage regimens and geriatric subjects.

### 2.4. Intravenous Pharmacokinetics

The model was initially optimized using data provided by the intravenous (IV) infusion pharmacokinetic study reported by Azzaro et al. [3,11]. The IV clearance input values (CL_iv_) of the parent and metabolites (except DMS) were obtained from the literature [3,36,37]. The input values for renal clearance values (CL_R_) of SEL and its metabolites were also obtained from the literature [3,36,37]. The parent exhibits protein binding to both macroglobulin and albumin; however, it preferentially binds to macroglobulin. In order to account for the binding to both proteins, the dissociation constants (K_D_) of SEL to macroglobulin and albumin were fitted to obtain the fraction unbound value reported in the literature. The K_D_ values of SEL to macroglobulin and albumin were fitted to 2.2 μM and 95 μM, respectively, which was based on the fact that SEL exhibits a stronger binding to macroglobulin compared to albumin [18,19]. The IV clearance value of DMS (CL_iv DMS_) was optimized based on parameter sensitivity analysis and a parameter estimation method, utilized in the explanation of observed pharmacokinetics of DMS. The full PBPK model explained the PK of the parent and major primary metabolite (MAP), whereas the minimal PBPK model was selected for DMS and AMP (secondary metabolite). The minimal PBPK for DMS and AMP was selected due to the limited capabilities of applying the full PBPK model to the minor primary metabolite and the secondary metabolite. Additionally, these compounds were closer to a volume of distribution value of 4 L/Kg and it is expected that application of the full PBPK to these metabolites will not improve the predicted outcome [38]. The tissue to plasma partition coefficients and the steady state volume of distribution were predicted based on the Rodgers and Rowland method (Method 2 in Simcyp) [39,40].

The metabolic conversion of parent to primary metabolites and further to the secondary metabolite were modeled based on the enzyme specific unbound intrinsic clearances (CL_u, int_). These CL_u, int_ values were calculated using the Simcyp retrograde reverse translational tool (RTT). Simcyp RTT calculates the unbound intrinsic clearance value of each CYP for a designated pathway using the hepatic well-stirred model and the corresponding absolute enzyme abundance as described in Equation (1). The mean intrinsic clearance values, used as the model input, were calculated from 1000 virtual simulations based on the age and gender distribution of the reported clinical trial [41]:(1)$${\mathrm{CL}}_{{\mathrm{intH}}_{\mathrm{Pathway}},x}\;\left(µL/\min/\mathrm{pmolEnzyme}\right)=\frac{\left(\frac{\%\;\mathrm{Hep}\;\mathrm{metCL}}{100}\right)\cdot \frac{Q_{H}\cdot \frac{\left({\mathrm{CL}}_{\mathrm{iv}}-{\mathrm{CL}}_{R}\right)}{B/P}}{\frac{\mathrm{fu}}{B/P}\cdot \left(Q_{H}-\frac{\left({\mathrm{CL}}_{\mathrm{iv}}-{\mathrm{CL}}_{R}\right)}{B/P}\right)}}{\mathrm{LW}\cdot \mathrm{MPPGL}\cdot \mathrm{CYPabundance}\cdot 60\cdot 10^{-6}}$$

The terms ${\mathrm{CL}}_{{\mathrm{intH}}_{\mathrm{Pathway}},x}$, fu, CL_iv_, CL_R_, B/P, Q_H_, % Hep metCL, LW, and MPPGL represent the intrinsic clearance of specific CYP enzymes involved in a designated pathway, fraction unbound, intravenous clearance, renal clearance, blood to plasma ratio, hepatic blood flow, percentage of relative contribution of each CYP (% Hep CL) to a designated metabolic pathway, liver weight, and microsomal protein per gram of liver, respectively. The % Hep CL values for SEL to MAP and DMS conversions, and the MAP to AMP conversion were obtained from the recombinant CYP 450 phenotyping data, which are summarized in Table 1 [15,26]. For the DMS to AMP conversion, the published studies, based on CYP reaction phenotyping by chemical inhibition (CYP 2B6: 38.4% at 0.5 μM, CYP 3A4: 23.5% at 5 μM and CYP 2A6: 8.7) and enzyme correlation (CYP 2B6: 0.9767, CYP 3A4: 0.9583 and CYP 2A6: 0.8272) data, suggest that the conversion is majorly mediated by CYP 2B6 and CYP 3A4 and to a minor extent by CYP 2A6 [26]. Therefore, due to a lack of experimentally measured fraction metabolized (fm) data for DMS to AMP conversion and based on the qualitative experimental findings, the fm values for CYP 2B6, CYP 3A4 and CYP 2A6 were assumed to be 0.45, 0.40, and 0.15, respectively. An intersystem extrapolation factor (ISEF) was applied to correct the enzyme intrinsic activity of isozymes (except for DMS to AMP conversion) relative to the human microsomes [42].

### 2.5. Transdermal Pharmacokinetics

Following the optimization of disposition characteristics through intravenous infusion PK, the transdermal absorption of the selegiline patch was evaluated mechanistically through the Multi-Phase Multi-Layer Mechanistic Dermal Absorption (MPML-MechDermA) model (Figure 2), which is integrated within a PBPK model. The detailed description of the MPML-MechDermA model has been provided elsewhere and the manuscript is currently under preparation [3,43]. The model has been verified against several compounds and different formulations [44,45,46,47,48]. Briefly, the model accounts for longitudinal diffusion and distribution across several layers of skin, which includes both transdermal and transappendageal pathways. The model delineates dermal specific physiological parameters such as tortuosity of the stratum corneum (SC) diffusion pathway, keratin protein binding, transappendageal hair follicular transport, pH at the surface of the skin, drug specific physicochemical properties and the formulation characteristics of topical applications (solutions, gels, emulsion creams, and suspensions), and transdermal patches that can be modeled through empirical release or through diffusion based kinetics. The stratum corneum, the rate limiting barrier, includes multiple layers of cuboidal corneocytes (with keratin and water) embedded in a lipid matrix, similar to a “brick and mortar” arrangement [46]. Other dermal layers, such as the viable epidermis, dermis, subcutis, and deep muscle, are currently modeled as single layered and well-stirred compartments. Blood flow to the dermis, subcutis, and muscle is modeled as a function of cardiac output. The model accounts for regional specific physiology (forehead, volar forearm, dorsal forearm, upper arm (dorsal part), face (cheek), lower leg, thigh, and back) and population-specific physiology differences (pediatrics, adults, and psoriatic patients). The transdermal absorption (diffusion and partition coefficients) across different layers of the skin were calculated through Quantitative Structure Activity Relationship (QSAR) of the permeants (drugs or chemicals). The QSAR describes the relationship between the physicochemical properties of the compound and the transdermal absorption parameters (partition and diffusion coefficients). The partition and diffusion coefficients and their corresponding QSARs, along with the equations, for SEL are summarized in Table 2.

The transdermal PK of the SEL patch were reported for various doses and dosage regimens and in various populations, which included healthy adults, geriatric populations, and hepatic and renal insufficiency patients (Appendix A). Transdermal PK was initially evaluated in a healthy subject single dose transdermal patch PK study through which the empirical release rate from the patch was optimized [3]. The optimized model was further verified against different doses and dosage regimens in healthy volunteers and the geriatric populations (Appendix A).

### 2.6. Prediction of Pharmacokinetics in Special Populations

The verified PBPK model was then utilized in the prediction of PK in adolescent (12–20 years old) and severe renal impairment populations at the therapeutic dose of 20 mg per 20 cm^2^. The trial simulations in renal and hepatic impaired subjects were performed in their corresponding Simcyp built-in populations as mentioned above. The simulation trials in these subjects matched the clinical trial design (number of subjects, age, gender, proportion of males etc.) reported in the EMSAM^®^ patch New Drug Approval package [11]. The simulation trials in the adolescent population were performed in 100 virtual subjects (10 trials and 10 subjects) using the Simcyp built-in Paediatric population [56,57].

### 2.7. Model Evaluation

Model performance was assessed by overlaying the simulated plasma concentration time profile (including the 5th and 95th percentiles) with the observed profile (mean ± SD), and where available, the variability in clinical PK profiles were included as error bars in the results. However, in most of the clinical PK studies, only the mean plasma concentration time profiles were reported with the exception of the transdermal PK study reported by Barret et al. [9]. The predictability of the PK parameters, area under curve (AUC), and the peak plasma concentration (C_max_) were evaluated by comparing their mean ratio ± SD ratio with the lines of identity (predicted/observed = 1) and were considered acceptable if they were within a 2-fold error (0.5–2-fold). The reported pharmacokinetic parameters across different studies for the same drug and formulation are associated with inter- study variability. Inter-study variability in the clinical parameters may bias the assessment of IVIVE prediction accuracy when predicted PK parameters are compared with data from one specific PK study [58]. Industry-wide publication on PBPK model qualification and reporting has described the reasoning for 2-fold acceptance criteria [59]. For the evaluation of accuracy and acceptability of predictions, a commonly applied criterion is for values to be within 2-fold of the observed values, since the results from one controlled clinical study may not be representative of the larger population, especially for drugs that exhibit high variability in PK or if the study sample size is small. Further discussion on the relevance of the accepted 2-fold criteria in the context of predictive modeling is discussed by Rostami-Hodjegan 2018 [60]. The selection of acceptance criteria for the PBPK model predictability of PK parameters was also reported by Abduljalil et al. [58]. The report suggested that the 2-fold prediction error is acceptable for most compounds but this varies for the drugs with high PK variability like midazolam and digoxin (2.5-fold is acceptable). The 2-fold criteria is acceptable for intermediate variability; however, tighter boundaries are required for compounds that have low PK variability (1.5-fold is acceptable). The mean ratio refers to the ratio of the predicted mean value to that of the observed value and the SD ratio which was calculated as described in Equation (2) [61]. The predicted mean value is the average of the predicted means across the 10 virtual trials and the predicted SD is the standard deviation of the predicted mean across the 10 virtual trials. The overall predictability of the model was evaluated in terms of precision, i.e., AAFE (average absolute fold error), as described in Equation (3), and bias, i.e., AFE (Average fold error) as described in Equation (4) [62,63]. The mean ratio ± SD ratios for the pharmacokinetic parameters were compared between the predicted parameters in special populations to that of healthy subjects:(2)$${\mathrm{SD}}_{\mathrm{ratio}}=\sqrt{{\left[\frac{\mathrm{SD}\left(\mathrm{observed}\right)}{\mathrm{Mean}\left(\mathrm{observed}\right)}\right]}^{2}+{\left[\frac{\mathrm{SD}\left(\mathrm{predicted}\right)}{\mathrm{Mean}\left(\mathrm{predicted}\right)}\right]}^{2}}\times \frac{\mathrm{Mean}\left(\mathrm{predicted}\right)}{\mathrm{Mean}\left(\mathrm{observed}\right)}$$
(3)$$\mathrm{AAFE}=10^{\frac{1}{N}\sum \left|\log\left(\frac{\mathrm{Predicted}}{\mathrm{Observed}}\right)\right|}$$
(4)$$\mathrm{AFE}=10^{\frac{1}{N}\sum \log\left(\frac{\mathrm{Predicted}}{\mathrm{Observed}}\right)}$$

## 3. Results

### 3.1. Model Development and Optimization

#### 3.1.1. Intravenous Pharmacokinetics

The PBPK model was initially developed and optimized against the 24-h IV infusion study (Dose—8.37 mg) reported by Azzaro et al. [3]. Overlays of the observed and simulated PK profiles of SEL and its metabolites are shown in Figure 3. The results demonstrated that the simulated PK profiles adequately reproduced the observed data within the 5th and 95th percentile range of the simulated data. The results in Figure 4a demonstrated that the mean and SD of the C_max_ and AUC_0-t_ ratio of SEL and its metabolites were within the acceptable limits of the prediction window. In addition, the model adequately recovered the percent fractions (%Fe) of SEL (Pred: 0.41 ± 0.05; Obs: 0.55 ± 0.31), MAP (Pred: 19.44 ± 3.84; Obs: 26.20 ± 6.55), DMS (Pred: 0.63 ± 0.42; Obs: 0.62 ± 0.25), and AMP (Pred: 6.33 ± 2.61; Obs: 11.08 ± 2.2) in urine [3,11]. The mean predicted metabolite to parent AUC_0-t_ ratio for MAP (Pred: 2.52; Obs: 2), DMS (Pred: 0.74; Obs: 0.51), and AMP (Pred: 0.9; Obs: 0.76) adequately captured the observed AUC_0-t_ metabolite to parent conversion ratio. Additionally, the optimized enzyme specific CL_u, int_ values for DMS adequately predicted the in vivo plasma PK profiles and the DMS/SEL conversion ratio. These findings relating to the intravenous infusion PK, indicate that the current PBPK model adequately captured the disposition characteristics of the parent and its metabolites.

#### 3.1.2. Single Dose Transdermal Pharmacokinetics in Healthy Male Subjects

The PK study reported by Azzaro et al. was utilized for the optimization of transdermal absorption. The simulations were performed as a 20 mg/20 cm^2^ single dose transdermal patch administered for 24 h in healthy male subjects (Appendix A) [3]. The release was more accurately explained by a slow first order release in comparison to a zero order or Higuchi release (Appendix A). The first order release rate was optimized by parameter sensitivity and estimation methods, the rate was found to be 0.04 h^−1^ by fitting transdermal PK data reported by Azzaro et al. (Appendix A) [3]. Although the in vitro release profiles of the SEL transdermal patch were reported, the data was not considered in the present study as it was not regarded as being a true representative of the in vivo profile. This was due to the fact that in the in vitro study, the% cumulative release was found to be 90% in 24 h, where, as in the in vivo scenario, only 6 mg was delivered in 24 h (for 20 mg/ 20 cm^2^, approximately 30% of the dose delivered in vivo) [5,26]. The SEL transdermal absorption related partition and diffusion coefficients are summarized in Table 3. Overlays of simulated mean pharmacokinetic profiles, with 5th and 95th percentiles, and the observed data are shown in Figure 3. It shows that the mean simulated PK profiles of SEL and its metabolites adequately explained the corresponding mean observed data. Additionally, the results in Figure 3 indicated that the ratio of mean and SD of C_max_ and AUC_0-t_ for SEL and its metabolites are within the acceptable limits of the prediction window. Furthermore, the model adequately recovered %Fe of SEL (Pred: 0.09 ± 0.02; Obs: 0.07 ± 0.056), MAP (Pred: 3.98 ± 1.40; Obs: 3.44 ± 1.24), DMS (Pred: 0.15 ± 0.12; Obs: 0.13 ± 0.09), and AMP (Pred: 1.31 ± 0.58; Obs: 1.39 ± 0.68) in urine [11]. The results of the plasma pharmacokinetics and urine excretion data of SEL and its metabolites suggest that the current PBPK model adequately captures the extent of transdermal absorption.

### 3.2. Model Verification

The PBPK model was further verified by simulating the single and multiple dermal doses reported in the literature [9,10,11]. Single dose PK profiles in adult healthy males (18.3 mg/10 cm^2^ for 24 h) presented in Figure 4a show that the model adequately predicted the individual observed pharmacokinetic profiles. The mean ± SD predicted to observed ratios were within the acceptable limits of the prediction window except for AUC_0-t_ DMS, which was due to a high variability in the observed AUC_0-t_ (68% CV) [10]. The single dose PK profiles in healthy elderly male and female subjects at 18.3 mg/10 cm^2^ presented in Figure 4c, show that model adequately predicted the observed pharmacokinetic profiles of SEL and its metabolites. The mean ± SD predicted to observed ratios of C_max_ and AUC_0-t_ for SEL (C_max_ ratio: 0.7 ± 0.4; AUC ratio: 0.7 ± 0.4 in males and C_max_ ratio: 0.8 ± 0.3 and AUC ratio: 0.7 ± 0.3 in females), MAP (C_max_ ratio: 0.6 ± 0.2; AUC ratio: 0.6 ± 0.2 in males and C_max_ ratio: 0.7 ± 0.4; AUC ratio: 0.7 ± 0.4 in females), DMS (C_max_ ratio: 0.5 ± 0.2; AUC ratio: 0.6 ± 0.3 in males and C_max_ ratio: 0.9 ± 0.6; AUC ratio: 0.9 ± 0.6 in females), and AMP (C_max_ ratio: 1.0 ± 0.5; AUC ratio: 1.1 ± 0.5 in males and C_max_ ratio: 1.0 ± 0.7; AUC ratio: 0.9 ± 0.5 in females) in males and females were within the acceptable limits of the prediction window (Figure 5e,f). Overall, the results from the predicted PK parameters indicated that there were no gender differences in the pharmacokinetics of SEL and its metabolites, which was in agreement with the observed clinical data (Appendix A) [9]. 

The multiple dose pharmacokinetics of SEL at therapeutic doses (20 mg and 30 mg per 20 cm^2^/24 h for 10 days) in elderly subjects were simulated using the Simcyp virtual geriatric population. The results of the multiple dose pharmacokinetic studies at these therapeutic doses showed that the mean predicted PK profiles adequately reproduced the observed mean PK data (Figure 4e,f). The mean ± SD predicted and observed ratios of steady state C_max_ and AUC_0-t_ for SEL and its metabolites are acceptable within the range of the prediction window. These results indicated that the model adequately explained the steady state pharmacokinetics of SEL and its metabolites at the recommended therapeutic doses. In the case of 7.5 mg per 5 cm^2^/24 h repeated dose for 10 days, the mean ± SD predicted and observed ratio of SEL AUC_0-t_ (1.9 ± 0.5) was at the boundary level, slightly above the acceptance limits (Figure 5g). The model predicted mean accumulation ratios (the AUC ratio at the last and first dose) of SEL (Pred: 3.67, Obs: 2.04), MAP (Pred: 25.83, Obs: 19.86), DMS (Pred: 12.40, Obs: 7.58), and AMP (Pred: 14.43, Obs: 11.75) for a repeated dose of 20 mg/20 cm^2^ per day for 10 days, matching the observed mean accumulation ratios. Furthermore, the predicted %Fe of SEL after 20 mg/20 cm^2^ per day for 10 days was found to be 0.72% which adequately captures the observed clinical percent excreted unchanged in urine (<1%) [11]. Overall, the model optimization and verification results indicated that the mean exposure ratio of SEL and its metabolites were within a two-fold error range of which seven of nine studies were within 0.67–1.5-fold range (50% error) and the remaining were within 0.5–2-fold range (Figure 5). The large variation in the AUC and C_max_ ratio was observed because of the high variability in the PK parameters reported in the clinical studies (% CV ≥ 30) in comparison to the mean predicted (mean of 10 virtual trials) and the standard deviation (standard deviation of the predicted mean across 10 trials). The mean ± SD of AUC and C_max_ for each trial and the observed mean ± SD are compared for further assessment of predicted versus observed variability (Appendix A). The results in Appendix A demonstrated that the model over predicted the variability of AUC and C_max_ for MAP for some of the clinical studies. This over prediction was due to the inclusion of poor metabolizers in the virtual clinical trial (8% of the population), in which CYP 2D6 is not expressed (Appendix A). To improve the predictions, CYP genotype information of the clinical data is required. Moreover, the model was able to capture the variability of the clinical data in the case of SEL, DMS, and AMP. These outcomes indicated that the observed PK parameters may be associated with high inter-individual variability, the number of subjects in the smaller clinical trials are not sufficient to represent the population mean and variability for the transdermal pharmacokinetics, and this was the reason for simulation of the clinical trial with 10 randomized virtual trials. In relevance to the present study, the clinically observed C_max_ (pg/mL) and AUC_0-24_ (h × pg/mL) values of SEL at the same transdermal dose of 20 mg/20 cm^2^ in healthy males demonstrates a high inter-study variability. Azzaro et al. (N = 13) [64] reported C_max_ (pg/mL) and AUC_0-24_ (h × pg/mL) values to be 2162.30 (78% CV) and 29425.87 (66% CV), respectively, and in another clinical study [11], C_max_ and AUC_0-24_ first dose values were reported to be 665 (33% CV) and 9520 (38% CV), respectively. In both these studies, the dose of SEL delivered was the same (6 mg/24 h). The difference in C_max_ (pg/mL) and AUC_0-24_ (h × pg/mL) values is more than 2-fold between clinical studies. Similarly, the inter-study variability was observed in another single transdermal PK study at 18.3 mg/10 cm^2^ in healthy males [9,10]. The overall bias (AFE) and precision (AAFE) for C_max_ of SEL (AFE: 0.87; AAFE: 1.33), MAP (AFE: 0.8; AAFE: 1.4), DMS (AFE: 0.89; AAFE: 1.36), and AMP (AFE: 1.22; AAFE: 1.29), and AUC_0-t_ of SEL (AFE: 1.02; AAFE: 1.44), MAP (AFE: 0.89; AAFE: 1.32), DMS (AFE: 1.15; AAFE: 1.38), and AMP (AFE: 1.20; AAFE: 1.29) were within 0.5–1.5-fold error [63].

### 3.3. Prediction of Pharmacokinetics in Special Populations

In order to understand the disposition differences associated with SEL, the comprehensive parent/metabolite PBPK model initially established in healthy adults and elderly subjects was further extrapolated to the special populations, specifically adolescents (12–20 years old) and in renal and hepatic insufficiency populations. The predicted PK parameters in these special populations were compared to the corresponding predicted parameters in healthy subjects, which were evaluated by the mean ± SD fold ratio (Figure 6). 

#### 3.3.1. Pharmacokinetics in Adolescents

The PK simulations of the selegiline transdermal patch in adolescent subjects (aged between 12–20 years) were performed as single (20 mg/20 cm^2^ for 24 h) and multiple (20 mg/20 cm^2^/24 h for 10 days) administrations. The predicted C_max_ and AUC_0-t_ parameters were compared between adolescents and adults using the mean ± SD fold ratio (Figure 6i,j). The single dose predicted C_max_ and AUC_0-t_ ratios of SEL and its metabolites, and also the steady state C_max_ and AUC_0-t_ ratios, were within the 0.67–1.5-fold ratio (50% error). The results indicated the pharmacokinetic simulations of SEL and its metabolites in adolescents did not exhibit any significant differences compared to healthy adult subjects.

#### 3.3.2. Prediction of Pharmacokinetics in Renally Impaired Subjects

To understand the differences between healthy and renally impaired subjects, the pharmacokinetic simulations of the selegiline transdermal patch in renally impaired subjects were performed as a single dose. Both total and free predicted C_max_ and AUC_0-t_ ratios were calculated to understand the differences between healthy and renally impaired subjects. The total and free C_max_ and AUC_0-t_ ratios of moderate renally impaired (MRI) subjects are summarized in Figure 6a,c, respectively. The mean total and free C_max_ and AUC_0-t_ ratios of SEL in MRI subjects were within 0.5–1.5-fold (Figure 6a,c) with about 40% CV through the mean ratio. The mean total free C_max_ and AUC_0-t_ ratios of MAP in MRI subjects were within 0.5–1.5-fold, with about 50% CV across the mean ratio (Figure 6a,c). The mean total C_max_ and AUC ratios of DMS in MRI subjects were found to be greater than 1.5-fold error and associated with higher variability (C_max_ ratio: 1.6 ± 1.3 and AUC ratio: 2.1 ± 2.0) as shown in Figure 6a. Similarly, the mean free C_max_ and AUC ratios of DMS were associated with greater than 1.5-fold error and associated with % CV > 80% across the mean ratio (Figure 6c). The mean exposure of AMP was reduced partially in comparison to healthy subjects, but it was associated with greater variability (>70% CV) Figure 6a,c.

The mean ± SD total and free C_max_, and AUC_0-t_ ratios of SEL and its metabolites in severe renally impaired (SRI) subjects are summarized in Figure 6b,d. The mean exposure ratio of free and total SEL was found to be within 0.5- to 1.5-fold and associated with about 40% CV across the mean ratio, similar to MRI subjects. The primary metabolite is associated with about 1.8-fold exposure ratio and associated with greater variability with percent CV > 70% (Figure 6b,d). These results indicated a higher systemic exposure of MAP in SRI subjects and a greater variability in its exposure (Figure 6b,d). The results, as shown in Figure 6b,d for DMS are associated with a mean exposure ratio greater than 2-fold and associated with variability of about 95% CV. The increase in the exposure of primary metabolites reduced the turnover of the secondary metabolite, which is also associated with higher variability in terms of exposure (Figure 6b,d). In summary, the simulations of the SEL patch indicated that there could be a significant increase in the systemic exposure of primary metabolites in MRI and SRI subjects and the two subject groups are associated with higher variability in terms of their exposure.

#### 3.3.3. Prediction of Pharmacokinetics in Hepatic Cirrhosis Subjects

Similar to MRI and SRI subjects, both total and free predicted C_max_ and AUC_0-t_ ratios were calculated to understand the difference between healthy and hepatic cirrhosis (Child-Pugh-B (CP-B) and Child-Pugh-C (CP-C)) subjects. The total and free C_max_ and AUC_0-t_ ratios of SEL and its metabolites in the CP-B population are summarized in Figure 6e,g, respectively. The mean total and free C_max_ and AUC_0-t_ ratios of SEL were found to be within 0.5- to 1.5-fold, with 40% CV, across the predicted mean ratio in CP-B subjects. The predicted C_max_ and AUC_0-t_ ratios of MAP in the CP-B subjects and the healthy subjects was found to be associated with a high variability range between 0.2–2.2. In the case of DMS the mean exposure ratio between the CP-B subjects and the healthy subjects was found to be within a range of 0.1–2.6. The exposure ratio of the secondary metabolite (AMP) was found to be in the range of 0.2- to 0.8-fold. 

In the case of the CP-C subjects, the total and free C_max_ and AUC_0-t_ ratios are summarized in Figure 6f,h, respectively. The total and free mean C_max_ and AUC_0-t_ ratios of SEL in the CP-C subjects and the healthy subjects were found to be within 0.5- to 1.5-fold with less than 40% CV across the mean ratio. In the case of MAP, the total and free mean C_max_ and AUC_0-t_ ratios between the CP-C subjects and the healthy subjects were found to be within the 0.2- to 2.2-fold range. In the case of DMS, the mean exposure ratio was found to be within the 0.1- to 1.9-fold range. The exposure variability of the primary metabolites resulted in a decreased turnover of the secondary metabolite (0–0.4) as shown in Figure 6f,h. In summary the outcomes of the pharmacokinetic simulations in healthy versus hepatic cirrhosis (CP-B and CP-C) subjects indicated that the systemic exposure of MAP and DMS were associated with higher variability.

### 3.4. Identification of Physiological Covariates

In order to further elucidate the differences and variability in the disposition of SEL and its metabolites between simulated healthy and special populations, the potential physiological covariates that are responsible for these differences were identified. The fraction-unbound values of SEL were found to be similar between healthy and renally impaired subjects (Figure 7). The reason for this could be due to the compensation effect of increased and decreased protein concentrations of macroglobulin and albumin, respectively [32,35]. Conversely, in the case of CP-C subjects, a slight increase in the fraction unbound value of SEL was observed; however, this did not affect the free concentrations of the parent. In SRI subjects, increased AUC_0-t_ and C_max_ values of MAP were observed when compared to the healthy subjects (Figure 6b and Figure 7). This increase was due to a decrease in CYP 2D6 abundance in MRI and SRI subjects in comparison to healthy subjects (Figure 7). The observed variability in the enzyme abundancies in healthy and renally impaired subjects illustrates the variability in the AUC_0-t_ and C_max_ values. On the other hand, the observed variability and decreased abundance of CYP 2B6 and CYP 3A4 in the renally impaired subjects explains the probable reason for the increased C_max_ and AUC_0-t_ values associated with DMS (Figure 7) in MRI and SRI subjects relative to the healthy subjects. The high variability in the exposure of MAP in CP-B and CP-C subjects was due to a decreased abundance of CYP 2D6 in these renally impaired subjects in comparison to healthy subjects (Figure 7). Similarly, a decrease in the abundance of CYP 2B6 and CYP 3A4 resulted in a high variability in exposure of DMS in the CP-B and CP-C subjects (Figure 7). Furthermore, the increased exposure of primary metabolism of SEL resulted in a decreased exposure of the secondary metabolite in renally and hepatic impaired subjects. 

## 4. Discussion

Special populations are often at high risk of sub- or supra-therapeutic effects, and these populations are often restricted or underrepresented in clinical trials [32]. The primary objective of the current study was to evaluate the PK differences of SEL, and its metabolites, between healthy and special populations through model-based analysis. The primary reason for this evaluation in special populations is that reported clinical studies in renal and hepatic populations are limited and are unexplored in the case of adolescents [11]. The desired outcome of the current modeling work is to determine whether dosing a SEL patch to these special populations requires closer clinical monitoring of any potential untoward effects of the parent drug and its metabolites [65]. 

In the present study, a comprehensive PBPK model, with the inclusion of a mechanistic dermal absorption model, allowed the capture of the PK of the parent drug (SEL), its primary metabolites (MAP and DMS), and the secondary metabolite (AMP). The overall precision and bias of the model in pharmacokinetics prediction of SEL and its metabolites were within 0.5- to 1.5-fold error. After successful verification, the model was further extrapolated to include special populations, specifically adolescents, MRI, SRI, CP-B hepatic cirrhosis, and CP-C hepatic cirrhosis subjects, which was performed in silico. The PK extrapolation of SEL and its metabolites to these special populations was plausible since these virtual populations account for the physiological differences between healthy and special populations and exhibit interrelationships that form a physiological network [27]. Sensitivity of a particular PK parameter to a potential covariate depends on the balance between drug properties and the sensitivities to elements within the physiological parameter network [27]. 

In context of the present study, SEL is a highly protein-bound compound that binds to both macroglobulin and albumin. The simulations in different populations indicated that the PK of SEL is sensitive to binding to both of these plasma proteins (Figure 7). The literature suggests that there is an increase in macroglobulin and a decrease in albumin in the renally impaired subjects [32,35]. Additionally, these protein concentrations are also influenced by age and gender distributions [27]. Due to a lack of literature evidence, the age and gender distributions of macroglobulin in this study were assumed to be similar to that of the reported distributions of albumin. The developed model accounts for these differences in plasma protein concentrations between healthy and renally impaired subjects. The mean free fractions of SEL in MRI and SRI subjects were found to be similar to that of healthy subjects, which was due to the opposing effects of both the proteins in those special populations when compared to the healthy population (Figure 7). The individual variability in the free fraction was due to the age and gender distribution of the plasma protein concentrations [27]. Conversely, the free fractions were slightly increased in the case of metabolites due to the observed decrease in albumin, the principal binding protein for metabolites [17]. The results in Figure 7 demonstrated that the free fractions of SEL and its metabolites varied within 0.5- to 1.5-fold in the renally impaired population when compared to that of the healthy subjects indicating a negligible effect of the decrease in plasma proteins on the free systemic exposure of SEL in the renally impaired population, as observed in Figure 6. In the case of hepatic cirrhosis CP-B and CP-C subjects, the free fractions were slightly above the mean predicted value observed in healthy subjects due to the decreased albumin concentration and negligible increase in macroglobulin concentration (1.2-fold higher than the mean concentration in healthy subjects) [35]. However, this did not affect the free systemic exposure of SEL due to the fact that macroglobulin is a principal binding protein. These findings, related to the fraction unbound variations across different populations, reiterates the fact that sensitivity of a PK parameter to potential covariates depends on the balance between the drug properties (selectivity of SEL to macroglobulin over albumin) and the physiological parameters (protein concentrations in the population). 

Evidence presented in the literature suggests that renal impairment not only alters the kidney function and plasma proteins but also alters the hepatic drug metabolism, tissue distribution, blood flow, and accumulation of metabolites, and it can even alter the disposition of non-renally eliminated drugs [32,66,67,68,69]. The developed model accounts for these key physiological changes in the renally impaired population, which were elaborately discussed by Rowland et al. [32]. The model predicted hepatic clearance calculated through in vitro-in vivo extrapolation (IVIVE) and renal clearance values of parent and metabolites in different populations, the predictions are summarized in Table 3. The predicted hepatic intrinsic clearance of SEL in SRI subjects decreased 3-fold in comparison to the healthy subjects, which was probably due to the decreased abundance of CYP enzymes (Table 3, Appendix A) [32]. In contrast, the in vivo hepatic clearance predicted when using the well-stirred model in SRI subjects (66.19 L/h) was reduced to only 23% in comparison to the healthy subjects (86.11 L/h) as shown in Table 3 [70]. SEL is a high hepatic extraction ratio drug (systemic clearance closer to the hepatic blood flow (89.3 L/h)), indicating that the rate of elimination is perfusion-limited. [71]. In addition, there was a 23% reduction in cardiac output (268.19 L/h) observed in SRI subjects in comparison to healthy subjects (350.19 L/h), which affected the total hepatic blood flow (SRI: 71.82 L/h). The outcomes of the present study indicated that a reduction in cardiac output is a potential covariate that affected the pharmacokinetics of SEL, which was also reflected in terms of AUC and C_max_ estimates of healthy and SRI subjects (Figure 6).

In the case of MAP, a 3-fold reduction in both the mean predicted intrinsic clearance and the hepatic clearance was observed in SRI subjects (Table 3). This reduction in clearance may be attributed to the fact that MAP is a low hepatic extraction ratio compound (hepatic blood flow: 89.3 L/h; calculated extraction ratio: 0.23). For a low hepatic extraction ratio compound such as MAP (less protein bound), the clearance is dependent on its intrinsic clearance [71]. MAP is predominantly excreted by the renal route; therefore, due to the decreased renal function in SRI subjects, an approximate 6-fold reduction in renal clearance was observed in SRI subjects when compared to healthy subjects. (Table 3). The literature suggests that there is a decreased CYP 2D6 activity in renally impaired subjects [21,32]. Therefore, the high variability in MAP to AMP conversion observed could be due to the variability in CYP 2D6 abundance in the renally impaired subjects (Figure 6 and Figure 7, Appendix A). The decreased CYP 2D6 abundance in SRI subjects is the potential physiological covariate that increased MAP exposure. In addition, the reduced renal function also decreased the renal clearance of MAP. Similar to MAP, about a 2.8-fold reduction in intrinsic clearance and a 2-fold reduction in hepatic clearance was observed with DMS. DMS is a moderate hepatic extraction ratio compound (hepatic blood flow: 89.3 L/h; calculated extraction ratio: 0.4), the clearance of such compounds are dependent on the hepatic blood flow, fraction unbound and the intrinsic clearance [71]. Renal impairment not only decreases the CYP 2D6 abundance but also other CYP enzymes such as CYP 2B6 and CYP 3A4, which are the potential physiological covariates that increased DMS exposure (Figure 6) [32]. The high variability in DMS to AMP conversion was probably due to the variability in CYP 2B6 and CYP 3A4 abundance (Figure 6 and Figure 7, Appendix A). Furthermore, the increased turnover of primary metabolites (MAP and DMS) resulted in a decreased exposure of AMP (Figure 6). The results for MRI subjects can be interpreted in a similar context to that of SRI subjects.

In regards to CP-C subjects, about a 10-fold reduction in mean intrinsic hepatic clearance of SEL (2109 L/h) was observed in comparison to the healthy subjects (24217.92 L/h). As discussed above, the rate of elimination of this compound is mostly perfusion limited, which explains why the total hepatic clearance in CP-C subjects was only reduced by 33%. In the case of CP-C subjects, only a 14% fold reduction in cardiac output (CP-C: 299.67 L/h) was observed. The results indicated that cardiac output is the potential covariate that caused the C_max_ and AUC_0-t_ to vary between 0.5- to 1.5-fold in comparison to the healthy subjects (Figure 6). In the case of MAP, a significant reduction (20-fold) in intrinsic clearance and the total hepatic clearance (14-fold) was observed in the CP-C subjects compared to the healthy subjects (Table 3). However, the exposure ratio (AUC_0-t_ ratio) of MAP in the CP-C subjects varied between 0.2- to 2.2-fold compared to that of the healthy subjects. This observed fold decrease in exposure was less than the anticipated exposure ratio (low hepatic extraction ratio), which was probably due to a significant reduction in the MAP formation from the parent (total intrinsic clearance to MAP: 1761.55 L/h in healthy subjects and 1613.58 L/h in CP-C subjects). Additionally, the extent of renal impairment in CP-C subjects (0.47) was found to be less than the impairment in SRI subjects (0.19), indicating that there could be a loss of MAP through the renal route since this compound is predominantly renally excreted, which could be the probable reason for the MAP exposure in CP-C subjects (Table 3 and Figure 6). About a three-fold and two-fold reduction in the predicted intrinsic clearance and hepatic clearance of DMS, respectively, was observed for CP-C subjects when compared to the healthy subjects. Moreover, the variability of the DMS exposure ratio between CP-C subjects and healthy subjects ranged from 0.1 to 3.1-fold. DMS is a moderate hepatic extraction ratio compound, the clearance of which is dependent on potential physiological covariates, predominantly the hepatic blood flow, fraction unbound and the intrinsic clearance [71]. The high variability in DMS exposure could be due to the variability in abundance of the CYP enzymes involved in the metabolism of DMS (Appendix A). An increase in the turnover of primary metabolites, due to decreased hepatic enzyme abundance, resulted in the significant decrease in the turnover of AMP (Figure 6). AMP is a renally-excreted compound and the exposure ratio of AMP between SRI and healthy subjects (Figure 6) was found to be higher than the exposure ratio between CP-C and healthy subjects. The probable reason could be due to a significant decrease in renal function in the SRI subjects compared to that of CP-C subjects (Table 3). On the other hand, the model simulation outcomes in adolescents, aged between 12–20 years, indicated no significant differences between the pharmacokinetics of the parent compound and its metabolites in the healthy subjects and adolescents. The results in Table 3 indicated the parent compound and its metabolites did not show any differences in fraction unbound values between healthy adults and adolescents. In the case of SEL, due to a lack of literature evidence the macroglobulin protein concentrations in the adolescents were assumed to be the same as the concentrations in adults. This assumption needs to be further investigated for accurate and reliable prediction results. However, differences in the albumin concentrations associated with age were considered in the present model. Furthermore, the hepatic intrinsic clearance and the hepatic clearance of the parent compound and its metabolites were assumed to be similar between healthy subjects and adolescents (12–20 years). This was probably due to no significant differences in the observed CYP enzyme abundance and cardiac output in the two populations (Figure 7 and Table 3, Appendix A).

To further understand the effect of these physiological covariates on the steady state (SS) concentrations of the parent compound and its metabolites, simulations were performed at a repeated dosage of 20 mg/20 cm^2^/24 h for 10 days, reflecting the therapeutic dosage regimen of SEL (20 mg/20 cm^2^ or 30 mg/20 cm^2^ per day). The results presented in Figure 8 demonstrate a 1.5-fold (47%) increase in free and bound concentrations of SEL in SRI subjects compared to healthy subjects. An increased accumulation of the primary metabolites was observed in SRI and CP-C subjects, which consequently reduced the turnover of the secondary metabolite. To express this increased accumulation of primary metabolites, the accumulation ratios of SEL and its metabolites were calculated; the data are summarized in Table 4. SEL did not exhibit any significant accumulation of the primary metabolites in SRI and CP-C subjects (Table 4). The simulations in SRI subjects indicated a 3-fold and 2-fold increase in primary metabolites (MAP and DMS) and the secondary metabolite (AMP), respectively. The observed increase in the metabolites was probably due to the decreased abundance of the metabolizing enzymes, in conjunction with renal insufficiency in the case of the primary metabolites. Renal insufficiency increased the accumulation of the secondary metabolite (AMP) in the SRI subjects (Table 4). A 5-fold increase in the accumulation ratios of primary metabolites was observed in CP-C subjects due to the decreased enzyme abundance of drug metabolizing enzymes. A 3-fold increase in secondary metabolite formation was observed in CP-C subjects. In terms of pharmacological activity, SEL and DMS exhibit anti-depressant effects through MAO-B inhibition; in addition, the SEL metabolites (MAP and AMP) are well known for their CNS stimulatory and psychoactive effects [7,72,73,74]. Suicidal tendencies are the major risk factor associated with anti-depressant drug therapy and the untoward effects of MAP and AMP are well documented [65,73,75]. In the present study, the likely key physiological covariates that are responsible for the disposition differences between healthy and special populations for the SEL transdermal patch were identified using a PBPK modeling approach. The current work focused only on the transdermal patch. Inclusion of other types of formulations via alternative routes (e.g., oral) would be beneficial for further model verification and building more confidence in metabolite formation. 

Oral and transdermal selegiline possess some fundamental clinical PK differences when SEL undergoes extensive first pass metabolism. These pharmacokinetic differences between healthy and special populations are more pronounced with the oral administration of selegiline [28]. Although the differences are less pronounced with the transdermal patch, the variability and differences in PK between healthy and the special population still exists. Moreover, the therapeutic uses are also different for both dosage forms where the oral selegiline is prescribed for late stage Parkinson disease and the transdermal selegiline is prescribed for major depressive disorder. The present study reports a model-based approach to evaluate the pharmacokinetic differences of SEL and its metabolites with the transdermal patch administration. The potential physiological covariates that affect the pharmacokinetics of SEL and its metabolites were identified in silico. The sensitivity to a potential covariate depends on the balance between the sensitivities of the elements within a physiological network. The outcome of the present study emphasizes the importance of closer clinical monitoring of renally and hepatic impaired subjects, during transdermal patch administration, to track the accumulation of SEL and its metabolites to identify any adverse side effects associated with SEL, when administered as an anti-depressant during chronic once daily repeated dose therapy.

## 5. Conclusions

In summary, the current study reports a comprehensive parent and metabolite PBPK model of SEL along with its metabolites. The model demonstrated acceptable predictive performance in explaining the observed pharmacokinetics of SEL and its metabolites in healthy and elderly subjects. Upon successful verification, the model was further extrapolated to three special population groups, renally impaired, hepatic impaired, and adolescent subjects. The model was able to capture the disposition differences in silico using virtual populations. The present study reports that a decrease in cardiac output in renally impaired and hepatic cirrhosis subjects can influence the SEL exposure. The decrease in CYP 2D6 abundance and its associated variability in renally impaired subjects enhanced the MAP exposure. The DMS exposure was varied due to a decrease in the abundance of CYP 2B6 and CYP 3A4 in renally and hepatic impaired subjects. Increased exposure of the primary metabolites (MAP and DMS) decreased the exposure of the secondary metabolite (AMP) in renally and hepatic impaired subjects. Since the therapeutic dosage regimen of the SEL transdermal patch is a once daily repeated dose, the variability in the exposure of the drug and its metabolites could lead to untoward side effects in MDD patients with renal and hepatic impairment. The outcome of the present study highlights the importance of closer clinical monitoring of these subjects upon SEL patch administration to identify any adverse side effects, since the anti-depressant therapy is associated with an increased suicidality risk.

## Figures and Tables

**Figure 1 pharmaceutics-12-00942-f001:**
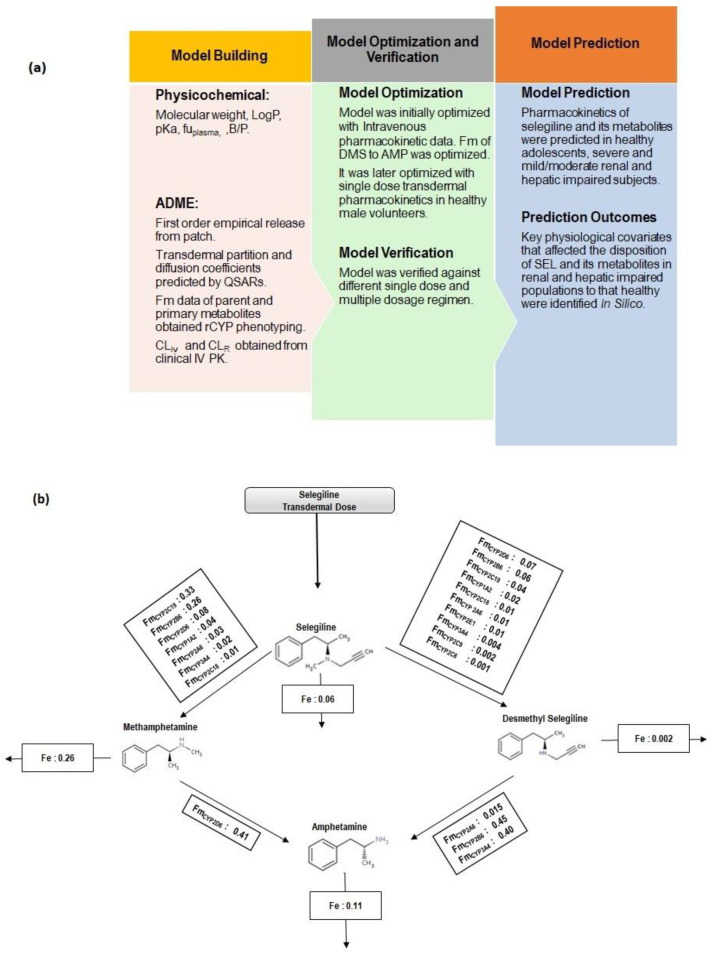
(**a**) Physiologically based pharmacokinetic model workflow. (**b**) Schematic representation of selegiline disposition along with Methamphetamine and Desmethyl Selegiline as primary metabolites and Amphetamine as a secondary metabolite. Transdermal absorption of selegiline was mechanistically modeled through the Multi-Phase Multi-Layer Mechanistic Dermal Absorption (MPML-MechDermA) model. Fm represents the fraction metabolized and Fe represents the fraction excreted unchanged in urine.

**Figure 2 pharmaceutics-12-00942-f002:**
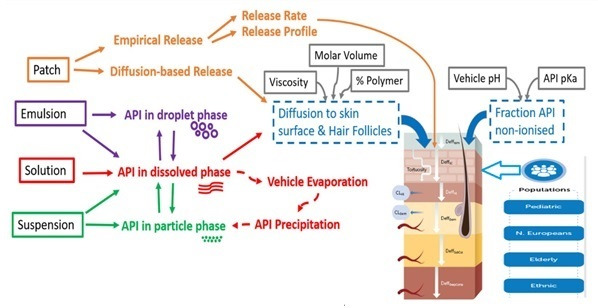
The Multi-Phase Multi-Layer Mechanistic Dermal Absorption (MPML-MechDermA) model.

**Figure 3 pharmaceutics-12-00942-f003:**
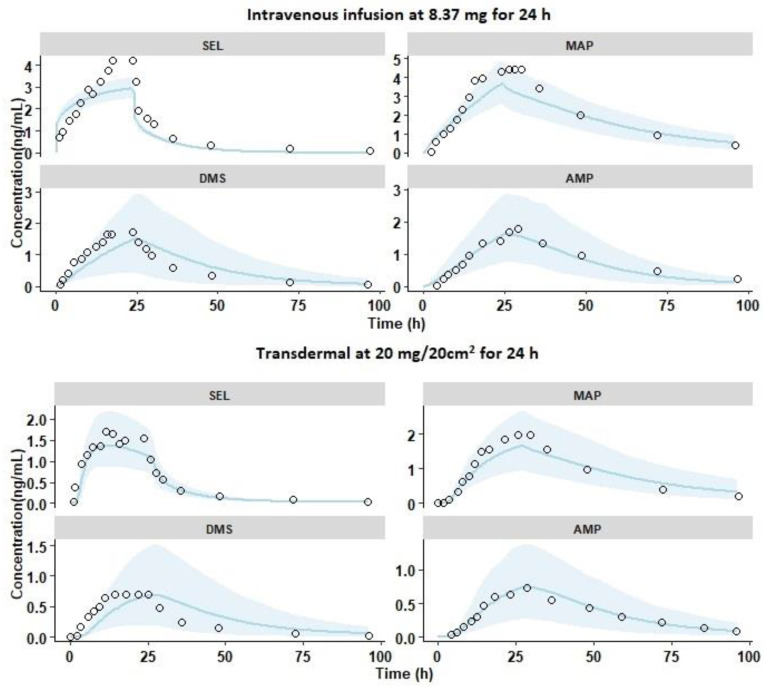
Overlay plots of observed and predicted mean pharmacokinetic profiles of SEL and its metabolites with 90% prediction interval represented as the shaded area (sky blue). The black open circles represent the mean observed data; the solid lines (sky blue) represent the simulated profiles. The compounds SEL: Selegiline, MAP: Methamphetamine, DMS: Desmethyl Selegiline, and AMP: Amphetamine are shown as faceted plots. The single dose intravenous infusion and transdermal administration of selegiline in healthy male subjects reported by Azzaro et al.

**Figure 4 pharmaceutics-12-00942-f004:**
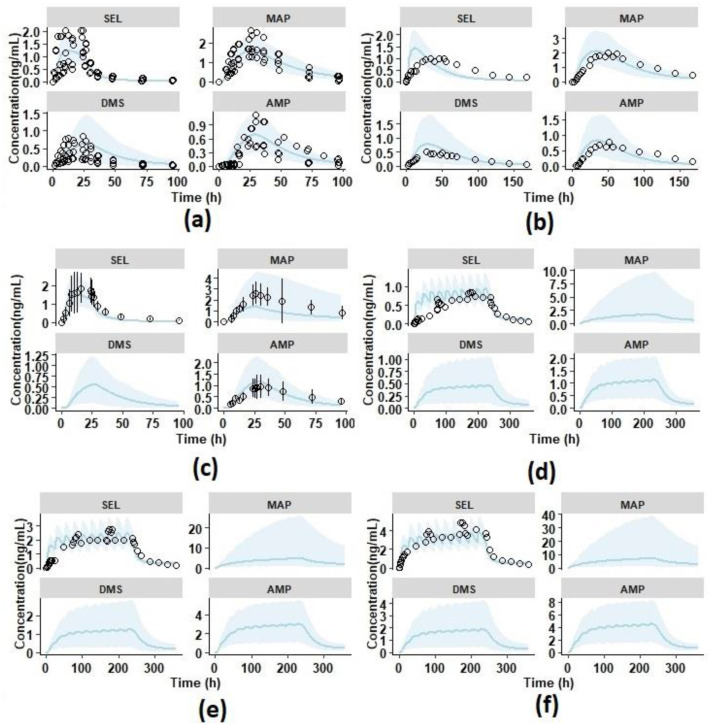
Overlay plots of observed and predicted mean pharmacokinetic profiles of SEL and its metabolites with 90% prediction interval represented as the shaded area (sky blue). The black open circles represent the mean observed data; the solid lines (sky blue) represent the simulated profiles. The compounds SEL: Selegiline, MAP: Methamphetamine, DMS: Desmethyl Selegiline, and AMP: Amphetamine are shown as faceted plots. (**a**) Single dose Transdermal pharmacokinetics in six healthy male adults at 18.3 mg/10 cm^2^ for 24 h with individual observed data; (**b**) single dose Transdermal pharmacokinetics in healthy male and female adults at 20 mg/20 cm^2^ for 168 h; (**c**) single dose Transdermal pharmacokinetics in healthy elderly subjects at 18.3 mg/10 cm^2^ for 24 h with mean ± SD observed data; (**d**) multiple dose Transdermal pharmacokinetics in healthy elderly subjects at 7.5 mg/5 cm^2^/24 h for 10 days; (**e**) multiple dose Transdermal pharmacokinetics in healthy elderly subjects at 20 mg/20 cm^2^/24 h for 10 days; (**f**) multiple dose Transdermal pharmacokinetics in healthy elderly subjects at 30 mg/20 cm^2^/24 h for 10 days.

**Figure 5 pharmaceutics-12-00942-f005:**
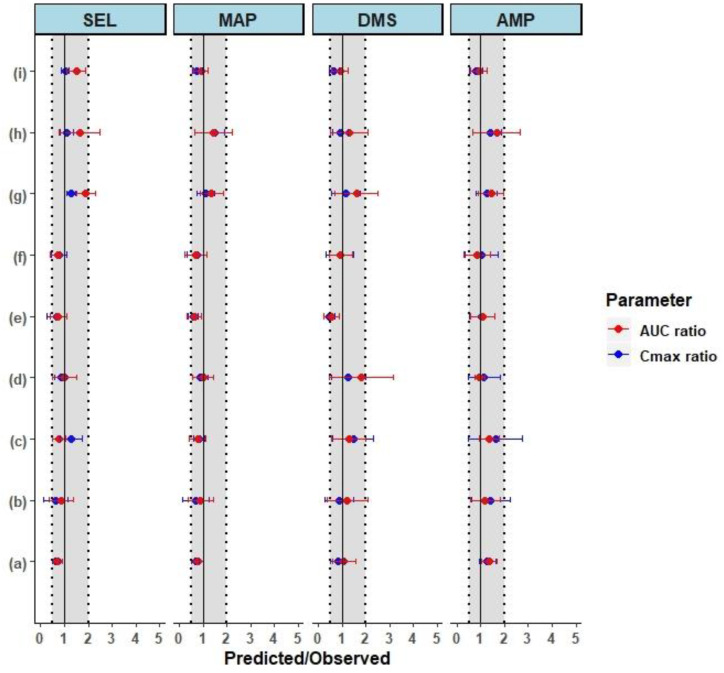
The forest plot shows the mean ± SD predicted in elderly subjects over the observed ratio of pharmacokinetic parameters in healthy subjects. The dotted and shaded area represents the 0.5- to 2-fold range and the solid black line represents the line of unity. The parameters AUC and C_max_ represent the Area Under the Curve 0-t and the maximum plasma concentration, respectively. The compounds SEL: Selegiline, MAP: Methamphetamine, DMS: Desmethyl Selegiline, and AMP: Amphetamine are shown as faceted plots. (**a**) Intravenous infusion at 8.37 mg/kg for 24 h; (**b**) single dose Transdermal pharmacokinetics in healthy male adults at 20 mg/20 cm^2^ for 24 h; (**c**) single dose Transdermal pharmacokinetics in healthy male and female adults at 20 mg/20 cm^2^ for 168 h; (**d**) single dose Transdermal pharmacokinetics in six male healthy adults at 18.3 mg/10 cm^2^ for 24 h; (**e**) single dose Transdermal pharmacokinetics in six healthy elderly males at 18.3 mg/10 cm^2^ for 24 h; (**f**) single dose Transdermal pharmacokinetics in six healthy elderly males and females at 18.3 mg/10 cm^2^ for 24 h; (**g**) multiple dose Transdermal pharmacokinetics in healthy elderly subjects at 7.5 mg/5 cm^2^/24 h for 10 days; (**h**) multiple dose Transdermal pharmacokinetics in healthy elderly subjects at 20 mg/20 cm^2^/24 h for 10 days; (**i**) multiple dose Transdermal pharmacokinetics in healthy elderly subjects at 30 mg/20 cm^2^/24 h for 10 days.

**Figure 6 pharmaceutics-12-00942-f006:**
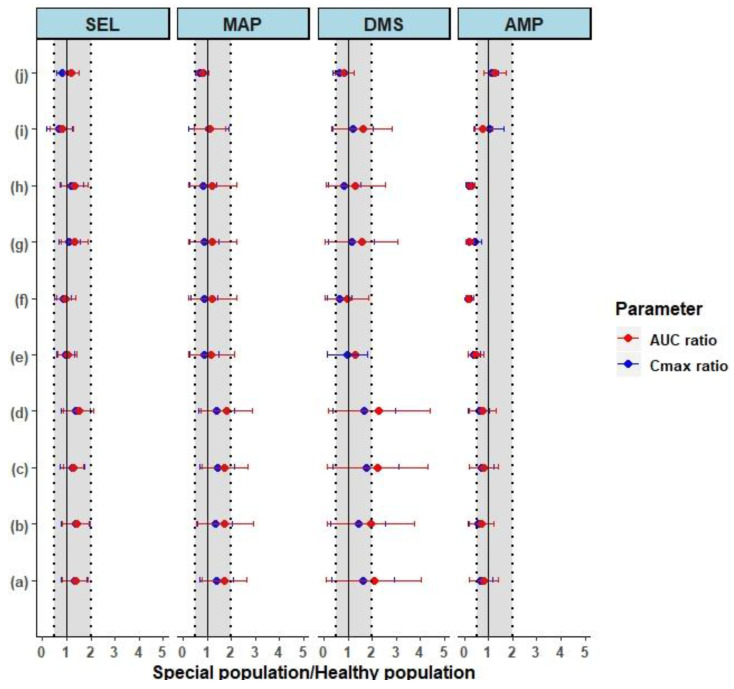
The forest plot shows the mean ± SD predicted in special populations over the predicted healthy ratio of pharmacokinetic parameters in healthy subjects. The dotted and shaded area represents the 0.5- to 2-fold range and the solid black line represents the line of unity. The compounds SEL: Selegiline, MAP: Methamphetamine, DMS: Desmethyl Selegiline, and AMP: Amphetamine are shown as faceted plots. (**a**) Single dose Transdermal pharmacokinetics in moderate renally impaired subjects versus healthy adults at 20 mg/20 cm^2^ for 24 h; (**b**) single dose Transdermal pharmacokinetics in severe renally impaired subjects versus healthy adults at 20 mg/20 cm^2^ for 24 h; (**c**) single dose Transdermal pharmacokinetics in moderate renally impaired subjects versus healthy adults with unbound pharmacokinetic parameters at 20 mg/20 cm^2^ for 24 h, (**d**) single dose Transdermal pharmacokinetics in severe renally impaired subjects versus healthy adults with unbound pharmacokinetic parameters at 20 mg/20 cm^2^ for 24 h; (**e**) single dose Transdermal pharmacokinetics in moderate Child-Pugh-B hepatic impaired subjects versus healthy adults at 20 mg/20 cm^2^ for 24 h; (**f**) single dose Transdermal pharmacokinetics in severe Child-Pugh-C hepatic impaired subjects versus healthy at 20 mg/20 cm^2^ for 24 h; (**g**) single dose Transdermal pharmacokinetics in moderate Child-Pugh-B hepatic impaired subjects versus healthy adults with unbound pharmacokinetic parameters at 20 mg/20 cm^2^ for 24 h; (**h**) single dose Transdermal pharmacokinetics in severe Child-Pugh-C hepatic impaired subjects versus healthy adults with unbound pharmacokinetic parameters at 20 mg/20 cm^2^ for 24 h; (**i**) single dose Transdermal pharmacokinetics in adolescent subjects versus healthy adults at 20 mg/20 cm^2^ for 24 h; (**j**) multiple dose Transdermal pharmacokinetics in adolescents versus adult subjects at 20 mg/20 cm^2^/24 h for 10 days.

**Figure 7 pharmaceutics-12-00942-f007:**
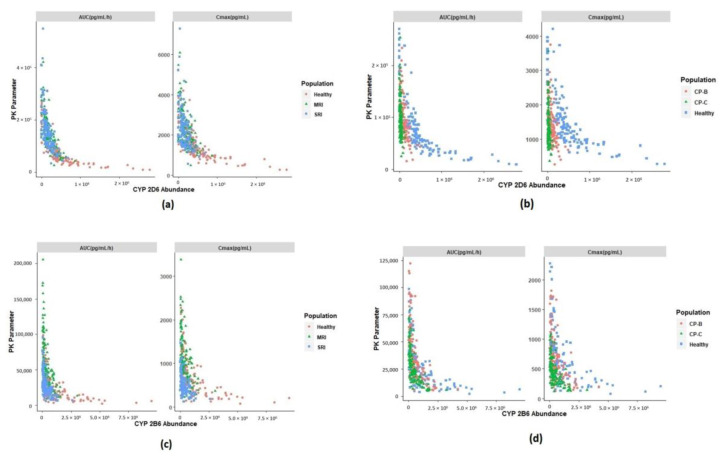
Potential physiological covariates that are responsible for disposition discrepancies between healthy and special populations. MRI: moderate renal impairment, SRI: Severe renal impairment, CP-B: Child Pugh-B hepatic cirrhosis, CP-C: Child Pugh-C hepatic cirrhosis. (**a**) PK parameters of Methamphetamine versus CYP 2D6 abundance (pmol/mg protein) in renal impairment subjects. (**b**) PK parameters of Methamphetamine versus CYP 2D6 abundance (pmol/mg protein) in hepatic cirrhosis subjects. (**c**) PK parameters of Desmethyl selegiline versus CYP 2B6 abundance (pmol/mg protein) in renal impairment subjects. (**d**) PK parameters of Desmethyl selegiline versus CYP 2B6 abundance (pmol/mg protein) in hepatic cirrhosis subjects. (**e**) PK parameters of Desmethyl selegiline versus CYP 3A4 abundance (pmol/mg protein) in renal impairment subjects. (**f**) PK parameters of Desmethyl selegiline versus CYP 3A4 abundance (pmol/mg protein) in hepatic cirrhosis subjects. (**g**) Fraction unbound plasma in different simulated populations.

**Figure 8 pharmaceutics-12-00942-f008:**
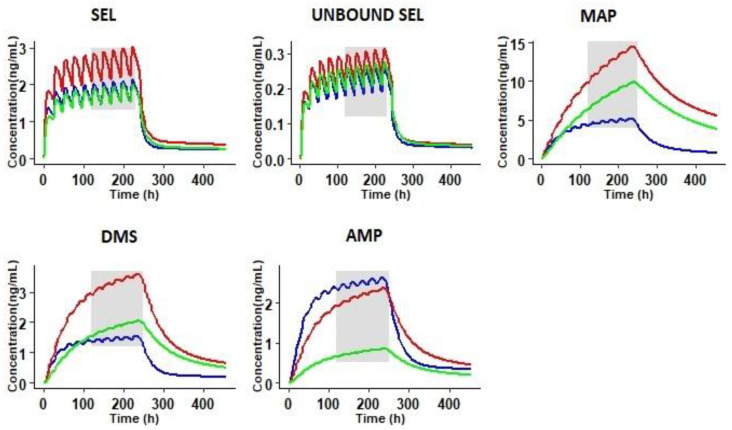
Steady state simulations at a repeated dose of 20 mg/20 cm^2^/24 h for 10 days in healthy (blue line), severe renally impaired (red line), and severe hepatic Child Pugh-C subjects (green line). The shaded area represents the steady state concentrations.

**Table 1 pharmaceutics-12-00942-t001:** Input parameters of SEL and its metabolites.

Parameter	Selegiline		Methamphetamine		Desmethyl Selegiline		Amphetamine
Molecular weight (g/mol) ^#^	187.28		149.23		173.25		135.2
Log P	2.7 ^a^		2.07 ^b^		2.46 ^c^		1.8 ^b^
Compound type	Monoprotic base		Monoprotic base		Monoprotic base		Monoprotic base
pKa	7.44 ^d^		9.9 ^e^		9.8 ^c^		9.9 ^e^
fu	0.1 ^f^		0.85 ^e^		0.47 *		0.84 ^e^
B/P	1.34 ^f^		1.487 ^$^		1.70 ^$^		1.35 ^$^
*Distribution*							
Distribution model	Full PBPK		Full PBPK		Minimal PBPK		Minimal PBPK
V_ss_ (L/Kg)	9.26		7.14		5.37		5.29
*Elimination*							
CL_iv_	84.56^g^		20 ^h^		35.8 *		20 *
CL_R_^g^	0.46		8.74		0.96		8.04
% Metabolized	1A2		MAP: 4.21		2D6		AMP:40
% Metabolized	1A2 2A6	DMS: 1.64	2D6	OHAMP:60	2A6	AMP: 15	
		MAP: 3.13		3A4	2B6	AMP: 45	
	2A6 2B6	DMS: 0.77				AMP: 40	
		MAP: 26.27					
	2B6 2C8	DMS: 6.48					
		DMS:0.10					
	2C9	DMS:0.23					
	2C18	MAP: 1.22					
	2C18 2C19	DMS: 1.05					
		MAP: 32.70					
	2C19 2D6	DMS: 3.58					
		MAP: 8.37					
	2D6 2E1	DMS: 7.20					
		DMS: 0.69					
	3A4	MAP:1.89					
	3A4	DMS:0.45					

SEL: Selegiline, DMS: Desmethyl selegiline, MAP: Methamphetamine, OHAMP: Hydroxy-amphetamine, AMP: Amphetamine. References: ^#^ (https://pubchem.ncbi.nlm.nih.gov/). ^a^ (Farahmand and Maibach 2009). ^b^ (Rezazadeh, Yamini et al. 2015) ^c^ (https://chemicalize.com). ^d^ (Völgyi, Ruiz et al. 2007). ^e^ (de la Torre, Farré et al. 2004). ^f^ (US FDA reference). * Optimized. ^$^ Predicted. ^g^ (Azzaro, Ziemniak et al. 2007). ^h^ (Mendelson, Jones et al. 1995). SEL: NDA:21-336/21-708 Clinical and Biopharmaceutics review, MAP: (Lin, Di Stefano et al. 1997), DMS: optimized.

**Table 2 pharmaceutics-12-00942-t002:** Absorption parameters.

Absorption Parameters	Parameter	Estimate	QSAR/Reference	Equation
Partition coefficients	Stratum corneum Lipid:vehicle	85.03	[49]	$K_{\mathrm{lip}:v}=1.32\times {\left(K_{o:w}\right)}^{0.67}{\;\mathrm{where}\;K}_{o:w}=10^{{\mathrm{LogP}}_{o:w}}$
	Sebum:vehicle	915.9	[50]	KSeb:v=10^((0.6044×LogPo:w)+1.33)
	Viable epidermis:Stratum corneum	24.8	[51]	$K_{\mathrm{SC}:\mathrm{VE}}=\frac{K_{\mathrm{lip}:v}}{0.7\times \left(0.68+\frac{0.32}{f_{u}}+\left(0.025\times f_{\mathrm{ni},\mathrm{VE}}\times K_{o:w}\right)\right)}$
	Viable epidermis:Sebum	0.09	Calculated	KVE:Seb=Klip;vKsb:v
	Dermis:Viable epidermis	1	Assumed	
	Subcutis:Dermis	1 × 10^−6^	Assumed	
	Dermis:blood	2.44	[52]	$K_{D:b}=\frac{0.98+0.02\times K_{\mathrm{ow}}}{0.9993+0.007\times K_{\mathrm{ow}}}$
	Subcutis:blood	1	Assumed	
Diffusion coefficients	Stratum corneum lipid	3.22 × 10^−5^	[53]	DSC,Lip(cm2h)=3600×(2×10−5×e−0.46×rc2)+3600×(3×10−9). r_c_: Molecular radius
	Viable epidermis	0.002	[51]	DVE(cm2h)=3600×DVE,free0.7×(0.68+0.32fu+(0.025×fni,VE×Klip:v)) ${\mathrm{where}\;D}_{\mathrm{VE},\mathrm{free}}\left(\frac{{\mathrm{cm}}^{2}}{\sec}\right)=10^{-4.38-\left(0.207\times {\mathrm{MW}}^{\frac{1}{3}}\right)}$
	Dermis	0.002	[51]	DD(cm2h)=3600×DVE,free0.7×(0.68+0.32fu+(0.025×fni,VE×Klip:v)) ${\mathrm{where}\;D}_{\mathrm{VE},\mathrm{free}}\left(\frac{{\mathrm{cm}}^{2}}{\sec}\right)=10^{-4.38-\left(0.207\times {\mathrm{MW}}^{\frac{1}{3}}\right)}$
	Sebum	0.0008	[54]	DSeb(cm2h)=3600×{A×MW−B+((KboltT4πηsbh)[ln(ηsbhηrc)−γe])} A = 0.000145; B = 1.32; Kbolt = 1.38 × 10^−16^ T [°K] = [°C] + 273.15 ($\eta_{\mathrm{sb}}\;\mathrm{is}\;\mathrm{the}\;\mathrm{viscosity}\;\mathrm{of}\;\mathrm{sebum}\;\mathrm{poise}\;\left(P\right)$
	Keratin Binding	Kon/koff78.8/0.93	[55]	${\mathrm{LogK}}_{b}=1.26+\left(0.34\times {\mathrm{LogD}}_{\mathrm{pH}}\right)$ $K_{\mathrm{off}}\left(h^{-1}\right)=\frac{60}{25.75+\left(8.35\times \left(D_{\mathrm{pH}}^{0.34}\right)\right)}{\;\mathrm{where}\;D}_{\mathrm{pH}}=10^{{\mathrm{LogD}}_{\mathrm{pH}}}$ Kon(h−1)=Koff×Kb where Kb=10^LogKb

QSAR: Quantitative structure activity relationship; K_lip;v_: Stratum corneum lipid:vehicle partition coefficient; K_o:w_: Octonal:water partition coefficient; K_Seb:v_: Sebum:vehicle partition coefficient; K_SC:VE_: Stratum corneum lipid:viable epidermis partition coefficient; K_VE:Seb_: Viable epidermis:Sebum partition coefficient; K_D:b_: Dermis:Blood partition coefficient; D_SC,Lip_: Stratum corneum lipid diffusion coefficient; D_VE_: Viable epidermis diffusion coefficient; f_u_: Fraction unbound; f_ni, VE_: Fraction unionized in viable epidermis; MW: Molecular weight; D_D_: Dermis diffusion coefficient; D_Seb_: Sebum diffusion coefficient; K_bolt_: Boltzman constant; K_b_: Equilibrium binding constant to keratin protein; K_on_: Association constant to keratin protein; K_off_: Dissociation constant to keratin protein.

**Table 3 pharmaceutics-12-00942-t003:** Hepatic and renal clearance of selegiline and its metabolites in different populations.

Population	Moiety	Hepatic Intrinsic Clearance (L/h)	Cardiac Output (L/h)	Hepatic Blood Flow (L/h)	Fraction Unbound (fu)	Renal Function ^a^	Hepatic Clearance (L/h)	Renal Clearance (L/h)
Healthy	SEL	24217.92	350.19	89.30	0.10	0.99	86.11	0.34
	MAP	27.70	350.19		1.00		21.14	8.6526
	DMS	125.44	350.19		0.47		35.36	0.9504
	AMP	24.745	350.19		0.84		16.79	7.9596
SRI	SEL	8221.1	268.79	71.82	0.10	0.16	66.19	0.05
	MAP	8.56	268.79		1.00		7.65	1.37
	DMS	45.31	268.79		0.55		18.40	0.15
	AMP	36.29	268.79		0.87		22.01	1.26
CP-C	SEL	2109.18	299.67	70.88	0.14	0.47	57.05	0.16
	MAP	1.45	299.67		1.00		1.42	4.05
	DMS	37.62	299.67		0.63		17.66	0.45
	AMP	30.11	299.67		0.91		19.70	3.73
Adolescents	SEL	20147.64	333.70	89.59	0.10	1.05	85.88	0.36
	MAP	20.89	333.70		1.00		16.94	9.11
	DMS	95.42	333.70		0.48		30.37	1.00
	AMP	17.05	333.70		0.84		12.40	8.38

SEL: Selegiline; MAP: Methamphetamine; DMS: Desmethyl Selegiline; AMP: Amphetamine. ^a^ Ratio of glomerular filtration rate (GFR) in a specific population compared to that of a normal GFR value (130 mL/min/1.73 m^2^ in males and 120 mL/min/1.73 m^2^ in females).

**Table 4 pharmaceutics-12-00942-t004:** The area under the curve (AUC) and accumulation ratios (AR) of SEL and its metabolites in healthy, severe renally impaired (SRI), and Child-Pugh-C hepatic cirrhosis (CP-C) subjects upon once daily repeated dose simulations of a SEL transdermal patch at 20 mg/20 cm^2^/24 h for 10 days.

Compound	Healthy			SRI			CP-C		
	AUC 0–24 h	AUC Last Dose	AR	AUC 0–24 h	AUC Last Dose	AR	AUC 0–24 h	AUC Last Dose	AR
SEL	25.70	113.30	4.41	34.60	174.39	5.04	21.54	121.91	5.66
MAP	21.75	479.48	22.05	29.34	2202.52	75.07	15.40	1577.26	102.39
DMS	8.14	115.26	14.15	9.47	384.40	40.58	3.81	258.65	67.82
AMP	10.94	210.16	19.21	5.61	258.96	46.18	1.67	102.29	61.28

SEL: Selegiline; MAP: Methamphetamine; DMS: Desmethyl Selegiline; AMP: Amphetamine; AR: Accumulation ratio.

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
