# Peer review of "Physiologically Based Pharmacokinetic Modeling of Transdermal Selegiline and Its Metabolites for the Evaluation of Disposition Differences between Healthy and Special Populations"

_pharmaceutics, 2020, doi:10.3390/pharmaceutics12100942_

Round 1

Reviewer 1 Report

I think the manuscript is initially impressive and contains interesting information. However, when comparing it with previous studies I see less novelty and fewer points where it is differentiated from previous studies, and also a lack of novelty in the purported clinical outcomes. In this last case I think the conclusions are very limited (i.e. monitoring of patients).

I am not sure the detailed study offers great improvements but, equally, there are no clear ground to refuse to publish this work. It seems, as I interpreted it, to require greater emphasis from the authors on why this work is novel and what benefits it can bring.

Thus, the novelty of the work needs to be clearly defined, as do the outcomes and conclusions. As they stand they really don't offer much of interest to a wider audience. 

Author Response

Reviewer 1

Open Review

English language and style

( ) Extensive editing of English language and style required 
(x) Moderate English changes required 
( ) English language and style are fine/minor spell check required 
( ) I don't feel qualified to judge about the English language and style 

Yes

Can be improved

Must be improved

Not applicable

Does the introduction provide sufficient background and include all relevant references?

( )

( )

(x)

( )

Is the research design appropriate?

( )

(x)

( )

( )

Are the methods adequately described?

( )

(x)

( )

( )

Are the results clearly presented?

( )

(x)

( )

( )

Are the conclusions supported by the results?

( )

( )

(x)

( )

Comments and Suggestions for Authors

Comment: I think the manuscript is initially impressive and contains interesting information. However, when comparing it with previous studies I see less novelty and fewer points where it is differentiated from previous studies, and also a lack of novelty in the purported clinical outcomes. In this last case, I think the conclusions are very limited (i.e. monitoring of patients).

I am not sure the detailed study offers great improvements but, equally, there are no clear ground to refuse to publish this work. It seems, as I interpreted it, to require greater emphasis from the authors on why this work is novel and what benefits it can bring.

Thus, the novelty of the work needs to be clearly defined, as do the outcomes and conclusions. As they stand, they really don't offer much of interest to a wider audience. 

Response: The authors would like to thank the reviewer for critical assessment of the manuscript and providing valuable feedback. The clinical pharmacokinetics of SEL along with its metabolites upon oral administration were reported in healthy, renal and hepatic impaired subjects [1]. However, the present study specifically focuses on SEL transdermal dosing, which is prescribed for major depressive disorder. Conversely, the oral selegiline tablet is prescribed as an adjunct therapy for late-Parkinson syndrome. Moreover, to the best of our knowledge, PBPK modelling of SEL and its metabolites have not been reported previously in scientific literature. SEL undergoes extensive first pass metabolism due to which the pharmacokinetic differences between healthy and special populations are more pronounced following oral selegiline administration [1]. The reference for the clinical pharmacokinetics of oral selegiline was added and discussed in the introduction section as per reviewer’s suggestion to clarify the difference between previous study and current manuscript. Please refer to Introduction section at lines 66-72 on page 2 The objective of the present study was to identify the key physiological covariates that are responsible for the disposition differences observed between healthy and special populations (adolescents, renal and hepatic impaired subjects) in silico using a comprehensive parent and metabolite physiologically based model upon administration of SEL patch. The simulations in adolescents populations indicated that no differences in the pharmacokinetics of parent and metabolites were observed when compared to the healthy subjects. The clinical pharmacokinetic studies of oral selegiline in renal and hepatic impaired subjects were reported in literature [1]. The pharmacokinetics oral selegiline is different from the transdermal selegiline due to the extensive first pass metabolism and the pharmacokinetics of parent are more affected to that of metabolites whereas the pharmacokinetics of metabolites are more affected in case of transdermal selegiline. The pharmacokinetic simulations of SEL patch along with its metabolites indicated that the PK differences are less pronounced upon transdermal patch when compared to the oral selegiline, however, the variability and differences in PK between healthy, renal and hepatic impaired subjects still exists.  In response to the reviewer’s suggestion, we have updated this in the discussion section. Please refer to lines 703-715 on page 22. The therapeutic uses are also different for both dosage forms where the oral selegiline is prescribed for late stage Parkinson disease and the transdermal selegiline is prescribe for major depressive disorder. The present study reports a model based approach to evaluate the pharmacokinetic differences of SEL and its metabolites upon the transdermal patch administration. The outcome of the present study highlights the importance of closer clinical monitoring of these subjects upon SEL patch administration to identify any adverse side effects, since the anti-depressant therapy is associated with an increased suicidality risk. The differences between the oral and transdermal selegiline therapy are described in the discussion section as per reviewer’s suggestion. Please refer to lines 703-715 on page 22.

Reference

  1. Anttila, M.; Sotaniemi, E.A.; Pelkonen, O.; Rautio, A. Marked effect of liver and kidney function on the pharmacokinetics of selegiline. Clinical Pharmacology & Therapeutics 2005, 77, 54-62.

Reviewer 2 Report

This manuscript describes the pharmacokinetic modeling based on physiological parameter of transdermal selegiline and its metabolites, comparing its disposition in healthy and special populations, where renal and hepatic impaired population including child is discussed.

The manuscript is nicely discussed and presented. I believe the study will be of high interest to attract wide readership and has clinical importance. I recommend this manuscript for publication.

Figure 3 & 4:  please include the details for color codes, lines, and data point labels, it will improve the readership. Some figures have error bars while other does not. Please explain

Author Response

Reviewer 2

Open Review

English language and style

( ) Extensive editing of English language and style required 
( ) Moderate English changes required 
(x) English language and style are fine/minor spell check required 
( ) I don't feel qualified to judge about the English language and style 

Yes

Can be improved

Must be improved

Not applicable

Does the introduction provide sufficient background and include all relevant references?

(x)

( )

( )

( )

Is the research design appropriate?

(x)

( )

( )

( )

Are the methods adequately described?

(x)

( )

( )

( )

Are the results clearly presented?

( )

(x)

( )

( )

Are the conclusions supported by the results?

(x)

( )

( )

( )

Comments and Suggestions for Authors

This manuscript describes the pharmacokinetic modeling based on physiological parameter of transdermal selegiline and its metabolites, comparing its disposition in healthy and special populations, where renal and hepatic impaired population including child is discussed.

The manuscript is nicely discussed and presented. I believe the study will be of high interest to attract wide readership and has clinical importance. I recommend this manuscript for publication.

Response: Authors would like to thank the reviewer for critical review of the work and encouraging comments.

Comment: Figure 3 & 4:  Please include the details for colour codes, lines, and data point labels, it will improve the readership.

Response: The black open circles represent the mean observed data; the solid line (sky blue) represents the simulated profile in the figure 3 and 4. The shaded area (sky blue) represents the 90 % prediction interval. This is now described in legends of the figures 3 and 4.

Comment: Some figures have error bars while other does not. Please explain

Response: Model performance was assessed by overlaying the simulated plasma concentration time profile (including the 5th and 95th percentiles) and the mean±SD observed PK profile. However, most clinical PK studies reported only the mean PK data, except the 18.3 mg/cm2 transdermal PK study, reported by Barret al 1996 [1]. The observed mean±SD PK profile was included for this report, indicated that the observed variability was within 5th and 95th percentiles of the simulations. To clarify this, a sentence is added in methods section “Where available, the variability in clinical PK profiles are included as error bars in the results. Please refer to .section 2.7 at lines 248-250 in page 9.

  1. Barrett, J.S.; Hochadel, T.J.; Morales, R.J.; Rohatagi, S.; DeWitt, K.E.; Watson, S.K.; DiSanto, A.R. Pharmacokinetics and safety of a selegiline transdermal system relative to single-dose oral administration in the elderly. American journal of therapeutics 1996, 3, 688-698.

Reviewer 3 Report

The authors have developed a PBPK model to characterize the PK time-course profiles of selegiline and its main primary (methamphetamine and desmethyl selegiline) and secondary metabolite (ampethamine). In general, the article sufficiently describes the methods and results obtained in a detailed manner. However, several concerns have been raised regarding the adequacy of the PBPK to be used as a valid tool for dose selection in special sub-groups of populations.

METHODS

-A more detailed description of the QSAR methodology used for the estimation of the transdermal absorption parameters is required.

-The acceptance limit of 2-fold error (0.5-2 fold) seems excessive during model development and optimization. Narrower limits should be considered.

RESULTS

-Model development started by using data from a 24 hour IV infusion study in a single dose level (8.37 mg). Based on results from Figure 3 and Figure 5a, the PBPK structure requires further improvement. The AUC and Cmax mean ratios for SEL and MAP are systematically under-predicted (including the SD) and AMP is systematically over-predicted (including the SD), showing that the model seems not able to adequately reproduce the observed behaviour. The mean predicted profile of SEL is not able to describe the experimental values (Figure 3). This phenomenon might suggest that the intrinsic clearance of SEL is over-predicted in order to be able to describe the primary (MAP and DMS) and secondary (AMP) metabolites. Therefore, the authors are encouraged to improve the PBPK.

-The authors are kindly requested to clarify wether the large variation on the AUC and Cmax ratios after transdermal administration of SEL (Figure 5b-h), based on the SD range, is derived from inter-variability or parameter uncertainty, since no sensitivity analysis was reported.

-Model predictions in special sub-groups of populations (pediatric patients and renally- and hepatic-impaired patients) should be considered with caution, based on the previous concerns raised. In addition, experimental information would be of great value in order to externally validate model predictions.

Author Response

Reviewer 3

Open Review

English language and style

( ) Extensive editing of English language and style required 
( ) Moderate English changes required 
( ) English language and style are fine/minor spell check required 
(x) I don't feel qualified to judge about the English language and style 

Yes

Can be improved

Must be improved

Not applicable

Does the introduction provide sufficient background and include all relevant references?

(x)

( )

( )

( )

Is the research design appropriate?

( )

(x)

( )

( )

Are the methods adequately described?

(x)

( )

( )

( )

Are the results clearly presented?

(x)

( )

( )

( )

Are the conclusions supported by the results?

( )

(x)

( )

( )

Comments and Suggestions for Authors

The authors have developed a PBPK model to characterize the PK time-course profiles of selegiline and its main primary (methamphetamine and desmethyl selegiline) and secondary metabolite (amphetamine). In general, the article sufficiently describes the methods and results obtained in a detailed manner. However, several concerns have been raised regarding the adequacy of the PBPK to be used as a valid tool for dose selection in special sub-groups of populations.

Response: Authors would like to thank the reviewer for their critical review and feedback, which has helped to clarify aspects regarding model inputs and performance verification for potential readers interested in modelling methodology and assessment.

Comment:  A more detailed description of the QSAR methodology used for the estimation of the transdermal absorption parameters is required.

Response: The QSAR describe the relationship between the physico-chemical properties of the compound and the transdermal absorption parameters (partition and diffusion coefficients). The QSARs for the transdermal absorption parameters are summarized in the table below. The QSAR equations are added to the table 2 in the manuscript including the references.

Parameter

Estimate

QSAR/Reference

Equation

Partition coefficients

Stratum corneum Lipid: vehicle

85.03

[1]

Sebum: vehicle

915.9

[2]

Viable epidermis: Stratum corneum

24.8

[3]

Viable epidermis: Sebum

0.09

Calculated

Dermis; Viable epidermis

1

Assumed

Subcutis:Dermis

1E-06

Assumed

Dermis: blood

2.44

[4]

Subcutis: blood

1

Assumed

Diffusion coefficients

Stratum corneum lipid

3.22E-05

[5]

. rc : Molecular radius

Viable epidermis

0.002

[3]

Dermis

0.002

[3]

Sebum

0.0008

[6]

A = 0.000145

B = 1.32

Kbolt = 1.38E-16

T [°K] = [°C] + 273.15 (

Keratin Binding

Kon/koff      78.8/0.93

[7]

Klip;v:Stratum corneum lipid:vehicle partition coefficient

Ko:w:Octonal:water partition coefficient

KSeb:v:Sebum:vehicle partition coefficient

KSC:VE:Stratum corneum lipid:viable epidermis partition coefficient

KVE:Seb:Viable epidermis:Sebum partition coefficient

KD:b:Dermis:Blood partition coefficient

DSC,Lip:Stratum corneum lipid diffusion coefficient

DVE: Viable epidermis diffusion coefficient

fu:Fraction unbound

fni, VE:Fraction unionized in viable epidermis

MW:Molecular weight

DD:Dermis diffusion ceoffcient

DSeb: Sebum diffusion coefficient

Kbolt:Boltzman constant

Kb:Equilibrium binding constantto keratin protein

Kon:Association constant to keratin protein

Koff:Dissociation constant to keratin protein

Comment The acceptance limit of 2-fold error (0.5-2 fold) seems excessive during model development and optimization. Narrower limits should be considered.

Response The reported pharmacokinetic parameters across different studies for the same drug and formulation are associated with significant inter- study variability. Inter-study variability in the clinical parameters may bias the assessment of IVIVE prediction accuracy when predicted PK parameters are compared with data from one specific PK study [8]. Industry-wide publication on PBPK model qualification and reporting has described the reasoning for 2-fold acceptance criteria [9]. “When evaluating the accuracy and acceptability of predictions, a commonly applied criteria is for values to be within 2-fold of the observed values. However, results from one controlled clinical study may not be representative of the larger population, especially for drugs that exhibit high variability in PK or if the sample size was small in such studies.” Further discussion on relevance of the used 2-fold criteria in the context of predictive modelling is discussed by Rostami-Hodjegan 2018 [10].The selection of acceptance criteria for the PBPK model predictability of PK parameters was also reported by Abduljalil et al [8,11]. The report suggested that the two fold prediction error is acceptable for most of the compounds but this varies for the drugs with high PK variability like midazolam and digoxin (2.5 can be acceptable), the 2 fold criteria is acceptable for intermediate variability and tighter boundaries are required for the compounds which have low PK variability (1.5 fold is acceptable). The description has been added to in methods section. Please refer to lines 269-287 and page 10.

In relevance to the present study, the observed Cmax (pg/mL) and AUC0-24 (h*pg/mL) values of SEL at same transdermal dose of 20 mg/20 cm2 in healthy males demonstrate high inter-study variability. Azzaro et al.(N=13) [12] reported Cmax (pg/mL) and AUC0-24 (h*pg/mL) to be 2162.30 (78 %CV) and 29425.87 (66 % CV) while in other clinical study [13], Cmax and AUC0-24 first dose were reported to be 665 (33 %) and 9520 (38 % CV). In both these studies, the amount delivered was same (6 mg/24 h). The difference in Cmax (pg/mL) and AUC0-24 (h*pg/mL) is more than 2-fold between these clinical studies. Similarly, the inter-study variability was observed for another single transdermal PK study at 18.3 mg/10 cm2 in healthy males [14,15].  Hence, the 2-fold criteria was used for SEL transdermal PBPK model prediction assessment. Nonetheless, the prediction errors were much smaller than 2-fold overall. The overall bias (AFE) and precision (AAFE) for Cmax of SEL (AFE:0.87; AAFE:1.33), MAP (AFE:0.8; AAFE:1.4), DMS (AFE:0.89; AAFE: 1.36),  and AMP (AFE:1.22; AAFE: 1.29),  and AUC0-t of SEL (AFE:1.02; AAFE:1.44), MAP (AFE:0.89; AAFE:1.32), DMS (AFE:1.15; AAFE:1.38), and AMP (AFE:1.20; AAFE:1.29) were within 0.5-1.5 fold error [16]. This is added in the manuscript. Please refer to lines 255-270 and page 10 and lines 367-376 and page 13.

Comment:  Model development started by using data from a 24-hour IV infusion study in a single dose level (8.37 mg). Based on results from Figure 3 and Figure 5a, the PBPK structure requires further improvement. The AUC and Cmax mean ratios for SEL and MAP are systematically under-predicted (including the SD) and AMP is systematically over-predicted (including the SD), showing that the model seems not able to adequately reproduce the observed behaviour. The mean predicted profile of SEL is not able to describe the experimental values (Figure 3). This phenomenon might suggest that the intrinsic clearance of SEL is over-predicted in order to be able to describe the primary (MAP and DMS) and secondary (AMP) metabolites. Therefore, the authors are encouraged to improve the PBPK.

Response:  The % Hep CL values for SEL to MAP and DMS conversions, and the MAP to AMP conversion were obtained from the recombinant CYP 450 phenotyping data summarized in clinical biopharmaceutics review of NDA 21-479 [17]. The CYP enzyme specific unbound intrinsic metabolic clearance of SEL to MAP and SEL to DMS were calculated using rearranged hepatic well-stirred model equation (Equation 1). The model predicted (well-stirred model) mean hepatic clearance of the parent calculated to be 84.22 L/h. The mean renal clearance of SEL was found to be 0.34 L/h and the model predicted mean total systemic clearance (Sum of hepatic and renal clearance) was estimated to be 84.56 L/h, which was closer to the observed systemic clearance (84.56 L/h). The results are summarized in Table 3 in the manuscript. The predicted AUC and Cmax ratios of SEL and its metabolites was within 0.6-1.5 fold ratio (50 % error) indicating the adequacy of the model in prediction of pharmacokinetics of SEL and its metabolites. In addition, the simulations were also performed using enzyme kinetics (Km and Vmax) results of SEL to MAP and SEL to DMS conversion, reported by Hidestrand et al [18]. The model adequately captured pharmacokinetics of the parent following intravenous infusion administration but under-predicted the MAP PK in terms of AUC and Cmax (Figure 1). The elimination slopes of observed and predicted metabolite were similar indicated the MAP (methamphetamine) to AMP (amphetamine) clearance is rightly captured but the turnover of MAP from SEL is less when Km and Vmax inputs used. Therefore, the retrograde model was used to capture the pharmacokinetics of the parent and metabolites. Further, the retrograde model calculates the mean intrinsic clearances from 1000 virtual subjects (as per equation 1) which has the age and gender distribution of the reported clinical trial.

Figure 1. Pharmacokinetics of SEL and MAP based on Km and Vmax reported Hidestrand et al.

Comment: The authors are kindly requested to clarify whether the large variation on the AUC and Cmax ratios after transdermal administration of SEL (Figure 5b-h), based on the SD range, is derived from inter-variability or parameter uncertainty, since no sensitivity analysis was reported.

Response: As described in section 2.7 lines 266-269 and page 9, the predictability of the PK parameters; Area under curve (AUC) and the peak plasma concentration (Cmax) were evaluated by comparing their mean ratio±SD ratio with the lines of identity (predicted/observed = 1) and were considered acceptable if they were within a 2 fold error. The mean ratio refers to the ratio of the predicted mean value to that of the observed mean value and the SD ratio, which was calculated as, mentioned in Equation 2. The predicted mean value is the average of the predicted means across the 10 virtual trials and the predicted SD is the standard deviation of the predicted mean across the 10 virtual trials. For further clarification in the manuscript, the details about calculation of the predicted mean and standard deviation is included in section 2.7 lines 272-273 and page 10. The observed data indicates that the parameters are associated with high variability that could be due to inter individual variability (% CV ≥30) when compared to the predicted standard deviation of mean of the 10 virtual trials. The mean±SD of AUC and Cmax of each of the 10 simulated trial and the observed mean±SD are now compared for further assessment of variability (Supplementary figure 3A and 3B, also provided below). These outcomes indicated that the observed PK parameters could be associated with high inter-individual variability, the number of subjects in the smaller clinical trials are not enough to represent the population mean and variability for the the transdermal pharmacokinetics, and this was the reason for simulation of the clinical trial with 10 randomized virtual trials. The supplementary figure 3A and 3B are added in the supplementary information of the revised manuscript. The results in figure 3A and 3B demonstrated that the model over predicted the variability of AUC and Cmax in case of MAP. The reason was due to inclusion of poor metabolizers in the virtual clinical trial (8% of populations), in whom CYP 2D6 is not expressed (Supplementary table2) and further improve the predictions, CYP genotype information of clinical data is required. Moreover, the model was able to capture the variability of the clinical data in case of SEL, DMS and AMP. Please refer to results section, line 381-386 and page12. Please refer to lines 376-398 and page 12.

Supplementary figure 3A. The plot represents the predicted mean±standard of maximum plasma concentration (Cmax) of 10 virtual trials and clinically observed mean±standard deviation. The observed data is represented in sky blue color and the predicted data is represented in dark blue color. The compounds SEL: Selegiline, MAP: Methamphetamine, DMS: Desmethyl Selegiline and AMP: Amphetamine are shown as faceted plots. (a) Intravenous infusion at 8.37 mg/kg for 24 hours, (b) Single dose Transdermal pharmacokinetics in healthy male adults at 20 mg/20 cm2 for 24 hours, (c) Single dose Transdermal pharmacokinetics in healthy male and female adults at 20 mg/20 cm2 for 168 hours, (d) Single dose Transdermal pharmacokinetics in six male healthy adults at 18.3 mg/10 cm2 for 24 hours, (e) Single dose Transdermal pharmacokinetics in six male healthy elderly males  at 18.3 mg/10 cm2 for 24 hours, (f) Single dose Transdermal pharmacokinetics in six male healthy elderly females  at 18.3 mg/10 cm2 for 24 hours, (g) Multiple dose Transdermal pharmacokinetics in healthy elderly subjects at 7.5 mg/5 cm2/24 hours for 10 days, (h) Multiple dose Transdermal pharmacokinetics in healthy elderly subjects at 20 mg/20 cm2/24 hours for 10 days, (i) Multiple dose Transdermal pharmacokinetics in healthy elderly subjects at 30 mg/20 cm2/24 hours for 10 days.

Supplementary figure 3B. The plot represents the predicted mean±standard of Area under the curve (AUC0-t) of 10 virtual trials and clinically observed mean±standard deviation. The observed data is represented in sky blue color and the predicted data is represented in dark blue color. The compounds SEL: Selegiline, MAP: Methamphetamine, DMS: Desmethyl Selegiline and AMP: Amphetamine are shown as faceted plots. (a) Intravenous infusion at 8.37 mg/kg for 24 hours, (b) Single dose Transdermal pharmacokinetics in healthy male adults at 20 mg/20 cm2 for 24 hours, (c) Single dose Transdermal pharmacokinetics in healthy male and female adults at 20 mg/20 cm2 for 168 hours, (d) Single dose Transdermal pharmacokinetics in six male healthy adults at 18.3 mg/10 cm2 for 24 hours, (e) Single dose Transdermal pharmacokinetics in six male healthy elderly males  at 18.3 mg/10 cm2 for 24 hours, (f) Single dose Transdermal pharmacokinetics in six male healthy elderly females  at 18.3 mg/10 cm2 for 24 hours, (g) Multiple dose Transdermal pharmacokinetics in healthy elderly subjects at 7.5 mg/5 cm2/24 hours for 10 days, (h) Multiple dose Transdermal pharmacokinetics in healthy elderly subjects at 20 mg/20 cm2/24 hours for 10 days, (i) Multiple dose Transdermal pharmacokinetics in healthy elderly subjects at 30 mg/20 cm2/24 hours for 10 days

Comment: Model predictions in special sub-groups of populations (paediatric patients and renally- and hepatic-impaired patients) should be considered with caution, based on the previous concerns raised. In addition, experimental information would be of great value in order to externally validate model predictions.

Response: The authors completely agree with reviewer’s comments that it is ideal to externally validate the model predictions for special populations. However, it is generally the case that such data is not available in literature. In absence of clinical studies in special population administered with SEL patch, the current work of using PBPK model provided mechanistically plausible estimation of likely parent and metabolite(s) disposition in special populations based on known differences in their physiology. When clinical data of SEL patch in special populations become available, authors would be able to include them in external validation of the model.

References

  1. Hansen, S.; Lehr, C.-M.; Schaefer, U.F. Improved input parameters for diffusion models of skin absorption. Advanced Drug Delivery Reviews 2013, 65, 251-264.
  2. Valiveti, S.; Wesley, J.; Lu, G.W. Investigation of drug partition property in artificial sebum. International Journal of Pharmaceutics 2008, 346, 10-16.
  3. Chen, L.; Han, L.; Saib, O.; Lian, G. In silico prediction of percutaneous absorption and disposition kinetics of chemicals. Pharmaceutical Research 2015, 32, 1779-1793.
  4. Shatkin, J.A.; Brown, H.S. Pharmacokinetics of the dermal route of exposure to volatile organic chemicals in water: a computer simulation model. Environmental research 1991, 56, 90-108.
  5. Mitragotri, S. Modeling skin permeability to hydrophilic and hydrophobic solutes based on four permeation pathways. Journal of Controlled Release 2003, 86, 69-92.
  6. Johnson, M.E.; Blankschtein, D.; Langer, R. Evaluation of solute permeation through the stratum corneum: lateral bilayer diffusion as the primary transport mechanism. Journal of pharmaceutical sciences 1997, 86, 1162-1172.
  7. Seif, S.; Hansen, S. Measuring the stratum corneum reservoir: desorption kinetics from keratin. Journal of pharmaceutical sciences 2012, 101, 3718-3728.
  8. Abduljalil, K.; Cain, T.; Humphries, H.; Rostami-Hodjegan, A. Deciding on success criteria for predictability of pharmacokinetic parameters from in vitro studies: an analysis based on in vivo observations. Drug Metabolism and Disposition 2014, 42, 1478-1484.
  9. Shebley, M.; Sandhu, P.; Emami Riedmaier, A.; Jamei, M.; Narayanan, R.; Patel, A.; Peters, S.A.; Reddy, V.P.; Zheng, M.; de Zwart, L. Physiologically based pharmacokinetic model qualification and reporting procedures for regulatory submissions: a consortium perspective. Clinical Pharmacology & Therapeutics 2018, 104, 88-110.
  10. Rostami‐Hodjegan, A. Reverse translation in PBPK and QSP: going backwards in order to go forward with confidence. Clinical Pharmacology & Therapeutics 2018, 103, 224-232.
  11. Clinical pharmacology and Biopharmaceutics reviews; United States Food and Drug administation: 02/27/2006.
  12. Azzaro, A.J.; Ziemniak, J.; Kemper, E.; Campbell, B.J.; VanDenBerg, C. Pharmacokinetics and absolute bioavailability of selegiline following treatment of healthy subjects with the selegiline transdermal system (6 mg/24 h): a comparison with oral selegiline capsules. The Journal of Clinical Pharmacology 2007, 47, 1256-1267.
  13. New Drug Application number : 21-336/21-708 Clinical Pharmacology and Biopharmaceutics Review; 21-336/21-708; United States Food and Drug Administration: 2006.
  14. Barrett, J.S.; Hochadel, T.J.; Morales, R.J.; Rohatagi, S.; DeWitt, K.E.; Watson, S.K.; DiSanto, A.R. Pharmacokinetics and safety of a selegiline transdermal system relative to single-dose oral administration in the elderly. American journal of therapeutics 1996, 3, 688-698.
  15. Rohatagi, S.; Barrett, J.S.; Dewitt, K.E.; Morales, R.J. Integrated pharmacokinetic and metabolic modeling of selegiline and metabolites after transdermal administration. Biopharmaceutics & drug disposition 1997, 18, 567-584.
  16. Maharaj, A.R.; Wu, H.; Hornik, C.P.; Arrieta, A.; James, L.; Bhatt-Mehta, V.; Bradley, J.; Muller, W.J.; Al-Uzri, A.; Downes, K.J. Use of normalized prediction distribution errors for assessing population physiologically-based pharmacokinetic model adequacy. Journal of Pharmacokinetics and Pharmacodynamics 2020, 1-20.
  17. New Drug Application number : 21-479 Clinical Pharmacology and Biopharmaceutics Review; 21-479; United States Food and Drug Administration: 2006.
  18. Hidestrand, M.; Oscarson, M.; Salonen, J.S.; Nyman, L.; Pelkonen, O.; Turpeinen, M.; Ingelman-Sundberg, M. CYP2B6 and CYP2C19 as the major enzymes responsible for the metabolism of selegiline, a drug used in the treatment of Parkinson's disease, as revealed from experiments with recombinant enzymes. Drug Metabolism and Disposition 2001, 29, 1480-1484.

Round 2

Reviewer 1 Report

The authors have addressed some of the issues from the original manuscript. However, the points raised in the initial review remain, even though the authors responded to this with a detailed commentary. So the comments remain unanswered in many ways. I believe that the manuscript should focus on more than one formulation to address the underlying work, that the variance in the data is worrying and despite changes to the manuscript is not really addressed.

Author Response

Open Review

English language and style

(x) Extensive editing of English language and style required 
( ) Moderate English changes required 
( ) English language and style are fine/minor spell check required 
( ) I don't feel qualified to judge about the English language and style #

Yes

Can be improved

Must be improved

Not applicable

Does the introduction provide sufficient background and include all relevant references?

(x)

( )

( )

( )

Is the research design appropriate?

( )

(x)

( )

( )

Are the methods adequately described?

(x)

( )

( )

( )

Are the results clearly presented?

( )

(x)

( )

( )

Are the conclusions supported by the results?

( )

(x)

( )

( )

Comments and Suggestions for Authors

Comment: The authors have addressed some of the issues from the original manuscript. However, the points raised in the initial review remain, even though the authors responded to this with a detailed commentary. So the comments remain unanswered in many ways. I believe that the manuscript should focus on more than one formulation to address the underlying work, that the variance in the data is worrying and despite changes to the manuscript is not really addressed.

 Response: We would like to thank the reviewer for the critical assessment and appreciate the suggestion on verifying the model with more than one formulation. Apart from transdermal patch, other topical or other dermal selegiline formulations are not available in the market to the best of authors’ knowledge. The approved oral formulations are available as tablets and capsules and some clinical PK reports for oral solutions were available. As it was mentioned in the introduction section at lines 68-74 on page 2 and discussed in the initial review response, the clinical pharmacokinetics of oral and transdermal selegiline are associated with some differences especially in terms of the first pass effect. To model the oral pharmacokinetics of parent and metabolites information about the intestinal metabolism and permeability characteristics are needed. Additionally the oral formulation characteristics and its PBPK modelling will be required. However, the focus of our current work as stated in title and throughout the manuscript is transdermal selegiline hence modelling oral formulation is beyond the scope of current manuscript. As mentioned in the initial response the therapeutic uses for oral (Parkinson’s syndrome) and transdermal (anti-depressant) are different. The outcome of the present study highlights the importance of closer clinical monitoring of these subjects upon SEL patch administration as the anti-depressant therapy is associated with an increased suicidality risk. The PBPK modeling of SEL and its metabolites upon oral administration itself qualifies to be a separate case study, which we are planning to communicate in future. This is now described in the discussion section as well for further clarification (lines 707-709 and page 22).